# Niche-specific genome degradation and convergent evolution shaping *Staphylococcus aureus* adaptation during severe infections

Stefano G Giulieri[1,2,3], Romain Guérillot[1], Sebastian Duchene[1], Abderrahman Hachani[1], Diane Daniel[1,4], Torsten Seemann[4], Joshua S Davis[5,6], Steven YC Tong[6,7], Bernadette C Young[8], Daniel J Wilson[9], Timothy P Stinear[1]*, Benjamin P Howden[1,2,4]*

[1]Department of Microbiology and Immunology at the Peter Doherty Institute for Infection and Immunity, University of Melbourne, Melbourne, Australia; [2]Department of Infectious Diseases, Austin Health, Heidelberg, Australia; [3]Victorian Infectious Diseases Service, Royal Melbourne Hospital, Melbourne, Australia; [4]Microbiological Diagnostic Unit Public Health Laboratory, The University of Melbourne at the Doherty Institute for Infection and Immunity, Melbourne, Australia; [5]Department of Infectious Diseases, John Hunter Hospital, Newcastle, New South Wales, Australia; [6]Menzies School of Health Research, Charles Darwin University, Casuarina, Northern Territory, Australia; [7]Victorian Infectious Disease Service, Royal Melbourne Hospital, and University of Melbourne at the Peter Doherty Institute for Infection and Immunity, Melbourne, Australia; [8]Nuffield Department of medicine, Oxford, United Kingdom; [9]Big Data Institute, Nuffield Department of Population Health, Li Ka Shing Centre for Health Information and Discovery, Old Road Campus, University of Oxford, Oxford, United Kingdom

**\*For correspondence:**
tstinear@unimelb.edu.au (TPS);
bhowden@unimelb.edu.au (BPH)

**Competing interest:** The authors declare that no competing interests exist.

**Abstract** During severe infections, *Staphylococcus aureus* moves from its colonising sites to blood and tissues and is exposed to new selective pressures, thus, potentially driving adaptive evolution. Previous studies have shown the key role of the *agr* locus in *S. aureus* pathoadaptation; however, a more comprehensive characterisation of genetic signatures of bacterial adaptation may enable prediction of clinical outcomes and reveal new targets for treatment and prevention of these infections. Here, we measured adaptation using within-host evolution analysis of 2590 *S. aureus* genomes from 396 independent episodes of infection. By capturing a comprehensive repertoire of single nucleotide and structural genome variations, we found evidence of a distinctive evolutionary pattern within the infecting populations compared to colonising bacteria. These invasive strains had up to 20-fold enrichments for genome degradation signatures and displayed significantly convergent mutations in a distinctive set of genes, linked to antibiotic response and pathogenesis. In addition to *agr*-mediated adaptation, we identified non-canonical, genome-wide significant loci including *sucA-sucB* and *stp1*. The prevalence of adaptive changes increased with infection extent, emphasising the clinical significance of these signatures. These findings provide a high-resolution picture of the molecular changes when *S. aureus* transitions from colonisation to severe infection and may inform correlation of infection outcomes with adaptation signatures.

## Editor's evaluation

This study offers a comprehensive examination of *Staphylococcus aureus* evolution during infection. It provides a useful analysis of select genetic signatures during bacterial adaptation. A combination of multiple layers of genome annotation and point mutation variant detection compellingly supports the correlation of infection outcomes with adaptation signatures in *S. aureus*.

## Introduction

While *Staphylococcus aureus* is one of the most important human pathogens (*Tong et al., 2015*), its common interaction with the human host is colonisation, usually of the anterior nares (*Wertheim et al., 2005*). Comparatively, severe, life-threatening infections such as bacteraemia or osteomyelitis occur very rarely. This suggests that at the macro-evolutionary level, *S. aureus* is primarily adapted to its natural ecological niche (the nasal cavity) and to specific selective pressures arising in this environment, such as competition with the resident microbiota (*Krismer et al., 2017*). By contrast, during invasive infection, a new fitness trade-off needs to be achieved to adjust to environmental challenges that include innate and acquired immune responses (*Proctor, 2019*), high-dose antibiotics (*Kuehl et al., 2020*), and nutrient starvation (*Hood and Skaar, 2012*). These trade-offs could occur across three potentially distinctive dynamics of micro-evolution during colonisation and infection (within the colonising population, from colonising to invasive, and within the invasive population), leading to nose-adapted, early infection-adapted, and late infection-adapted strains. Identifying infection-adapted strains might assist precision medicine strategies for infection prevention and management and refine the understanding of *S. aureus* pathogenesis versatility, as mutational footprints of selection mirror functions that are critically important for bacterial survival during invasion.

Emerging genomic approaches for analysis of within-host evolution are among the most powerful means to study bacterial host adaptation (*Marvig et al., 2015*; *Kennemann et al., 2011*; *Lieberman et al., 2016*). Studies have shown the remarkable diversity and evolution of colonising populations of *Streptococcus pneumoniae* (*Chaguza et al., 2020*) and *S. aureus* (*Paterson et al., 2015*). In *S. aureus*, *Enterococcus faecalis* and *Enterococcus faecium* it has been shown that transition from colonisation to invasion favours strains with specific adaptive signatures (*Young et al., 2017*; *Van Tyne et al., 2019*; *Chilambi et al., 2020*), while evidence of niche adaptation was noted in a within-host study of bacterial meningitis due to *S. pneumoniae* (*Lees et al., 2017b*). Furthermore, phenotypic and genomic adaptation (often in response to antibiotic pressure) has been investigated during selected episodes of persistent invasive infections due to *S. aureus* (*Giulieri et al., 2020*; *Howden et al., 2006*; *Giulieri et al., 2018*), *Pseudomonas aeruginosa* (*Wheatley et al., 2021*), *Salmonella enterica* (*Klemm et al., 2016*), and *Mycobacterium tuberculosis* (*Vargas et al., 2021*). To increase power, bacterial within-host evolution studies have leveraged on large collections of paired samples coupled with statistical models of genome-wide mutation rates (*Marvig et al., 2015*; *Lees et al., 2017b*; *Gatt and Margalit, 2021*) and extended the analysis to include chromosomal structural variation (*Giulieri et al., 2018*; *Lees et al., 2017a*; *Gao et al., 2015*) as well as intergenic mutations (*Lees et al., 2017a*; *Khademi et al., 2019*).

Convergent evolution among separated (independent) episode of colonisation or infection is a key indication of adaptation in evolution analyses. However, with the notable exception of one study of cystic fibrosis (*Long et al., 2020*), the convergence has generally been weak in within-host studies of *S. aureus* infections, with no convergence at all (*Giulieri et al., 2018*) or significant enrichment limited to the *S. aureus* master regulator *agr* (*Young et al., 2017*). We hypothesised that in addition to the small sample size, the extended range of bacterial functions potentially under selective pressure (each function being potentially targeted by diverse pathoadaptive mutations) has hampered the identification of important adaptation mechanisms. To overcome the limitations of studies to date, we have pooled all publicly available *S. aureus* within-host evolution studies, and complemented this with a new dataset from a recent *S. aureus* clinical trial (*Tong et al., 2020*), in a single large-scale analysis. Rather than focussing on point mutations and small insertions/deletions alone, we leveraged multiple layers of genome annotation (encompassing coding regions, operons, intergenic regions, and functional categories) and included chromosome structural variants to compile a comprehensive catalogue of bacterial genetic variation arising during host infection. This strategy enabled the detection of convergent adaptation patterns at an unprecedented resolution. We also outline distinctive

**eLife digest** The bacterium *Staphylococcus aureus* lives harmlessly on our skin and noses. However, occasionally, it gets into our blood and internal organs, such as our bones and joints, where it causes severe, long-lasting infections that are difficult to treat.

Over time, *S. aureus* acquire characteristics that help them to adapt to different locations, such as transitioning from the nose to the blood, and avoid being killed by antibiotics. Previous studies have identified changes, or 'mutations', in genes that are likely to play an important role in this evolutionary process. One of these genes, called accessory gene regulator (or *agr* for short), has been shown to control the mechanisms *S. aureus* use to infect cells and disseminate in the body. However, it is unclear if there are changes in other genes that also help *S. aureus* adapt to life inside the human body.

To help resolve this mystery, Giulieri et al. collected 2,500 samples of *S. aureus* from almost 400 people. This included bacteria harmlessly living on the skin or in the nose, as well as strains that caused an infection. Gene sequencing revealed a small number of genes, referred to as 'adaptive genes', that often acquire mutations during infection. Of these, *agr* was the most commonly altered. However, mutations in less well-known genes were also identified: some of these genes are related to resistance to antibiotics, while others are involved in chemical processes that help the bacteria to process nutrients.

Most mutations were caused by random errors being introduced in to the bacteria's genetic code which stopped genes from working. However, in some cases, genes were turned off by small fragments of DNA moving around and inserting themselves into different parts of the genome.

This study highlights a group of genes that help *S. aureus* to thrive inside the body and cause severe and prolonged infections. If these results can be confirmed, it may help to guide which antibiotics are used to treat different infections. Furthermore, understanding which genes are important for infection could lead to new strategies for eliminating this dangerous bacterium.

signatures of adaptation during colonisation, upon transition from colonisation to infection and during invasive infection.

## Results

### The *S. aureus* within-host evolution analysis framework

We compiled a collection of 2251 *S. aureus* genomes from 267 independent episodes of colonisation and/or infection, reported in 24 genomic studies (*Young et al., 2017*; *Giulieri et al., 2018*; *Gao et al., 2015*; *Young et al., 2012*; *Wuthrich et al., 2019*; *Trouillet-Assant et al., 2016*; *Tan et al., 2019*; *Suligoy et al., 2018*; *Rouard et al., 2018*; *Rishishwar et al., 2016*; *Petrovic Fabijan et al., 2020*; *Miller et al., 2020*; *Loss et al., 2019*; *Liu et al., 2020*; *Langhanki et al., 2018*; *Kuroda et al., 2019*; *Ji et al., 2020*; *Howden et al., 2011*; *Harkins et al., 2018*; *Golubchik et al., 2013*; *Burd et al., 2014*; *Benoit et al., 2018*; *Azarian et al., 2019*; *Altman et al., 2018*; *Table 1*; *Table 1—source data 1*). We supplemented this dataset of publicly available sequences with unpublished sequences from 603 serial invasive isolates collected within the CAMERA-2 trial (*Tong et al., 2020*).

Using genetic distance and sequence type (ST) to define within-host lineages, we estimated that coinfection was present in 4/336 (1%) of invasive episodes and co-colonisation 11/167 (7%). We removed genetically unrelated strains within the same episode and included 2590 genomes (1397 invasive and 1193 colonising) from 396 episodes in our within-host evolution analysis (*Figure 1*, *Table 1*, *Supplementary files 1 and 2*). The most prevalent lineages in the collection were ST 30 (342 strains, 13%), ST 22 (277 strains, 11%), and ST 5 (271 strains, 11%); 1001 strains (39%) were *mecA* positive. The collection was representative of the global *S. aureus* diversity, with an even distribution of colonising and invasive strains across the major clades (*Figure 2A*). The most frequent infection syndrome was bacteraemia without focus (152 episodes, 38.4%), while nasal carriage (166 episodes, 42%) was the most prevalent colonisation condition (*Table 1*).

Our within-host evolution analysis strategy identified 4556 genetic variants (median 3 per episode, range 0–237) (*Supplementary file 3*). Importantly, by investigating both point mutations and structural variation, we were able to uncover 214 large deletions (≥500 bp), 160 new insertion sequences (IS)

**Table 1.** Microbiological and clinical characteristics of the colonisation and infection episodes included in the within-host evolution analysis.

*Table 1—source data 1* provides a list of the within-host studies included in the analysis.

| | Strains (n=2590) | Episodes (n=396) |
|---|---|---|
| **Sequence type** | | |
| 30 | 342 (13.2%) | 43 (10.9%) |
| 22 | 277 (10.7%) | 44 (11.1%) |
| 5 | 271 (10.5%) | 42 (10.6%) |
| 45 | 198 (7.6%) | 38 (9.6%) |
| 15 | 156 (6.0%) | 4 (3.5%) |
| 1 | 133 (5.1%) | 14 (3.5%) |
| 93 | 110 (4.2%) | 29 (7.3%) |
| 8 | 107 (4.1%) | 18 (4.5%) |
| 239 | 100 (3.9%) | 29 (7.3%) |
| Other | 896 (34.6%) | 125 (31.6%) |
| *mecA* positive | 1001 (38.6%) | 207 (52.3%) |
| Infection syndrome | | |
| Skin infection | 204 (7.9%) | 32 (8.1%) |
| Osteoarticular infection | 77 (3.0%) | 17 (4.3%) |
| Bacteraemia without focus | 588 (22.7%) | 152 (38.4%) |
| Bacteraemia with focus | 331 (12.8%) | 85 (21.5%) |
| Endocarditis | 197 (7.6%) | 44 (11.1%) |
| No invasive strains | | 66 (16.7%) |
| Colonisation syndrome | | |
| Nasal carriage | 974 (37.6%) | 166 (42%) |
| Cystic fibrosis | 57 (2.2%) | 9 (2%) |
| Atopic dermatitis | 162 (6.3%) | 9 (2%) |
| No colonising strains | | 212 (54%) |

The online version of this article includes the following source data for table 1:

**Source data 1.** List of within-host studies included in the analysis.

insertions, and 609 copy number variants, underscoring the role of large chromosome structural variation in within-host evolution. To increase the evolutionary convergence signal by aggregating mutations in functionally consistent categories, we annotated all genetic variants using multiple datasets, including coding sequences, regulatory intergenic regions, operons and gene ontologies (*Figure 1C*).

## Distinctive evolutionary patterns define nose-adapted, early infection-adapted, and late infection-adapted strains

Based on the working hypothesis that *S. aureus* host adaptation patterns differ according to whether the strains are nose-colonising, collected at an early stage of infection (i.e. within the first 3 days) or at a late infection stage (i.e. associated with persistence beyond the first 3 days or recurrence), we assessed whether it was possible to define (i) general paradigms of genetic variation and (ii) specific convergence signatures. Thus, we classified within-host acquired variants into three groups according to their most likely location in the within-host phylogeny: (i) between colonising strains (colonising-colonising [type C>C]); (ii) between colonising and early infection adapted strains (colonising-invasive [type C>I]); and (iii) between invasive strains (invasive-invasive [type I>I]). Overall, the 396 infection episodes included in the analysis allowed us to independently assess 166 type C>C, 118 type C>I, and

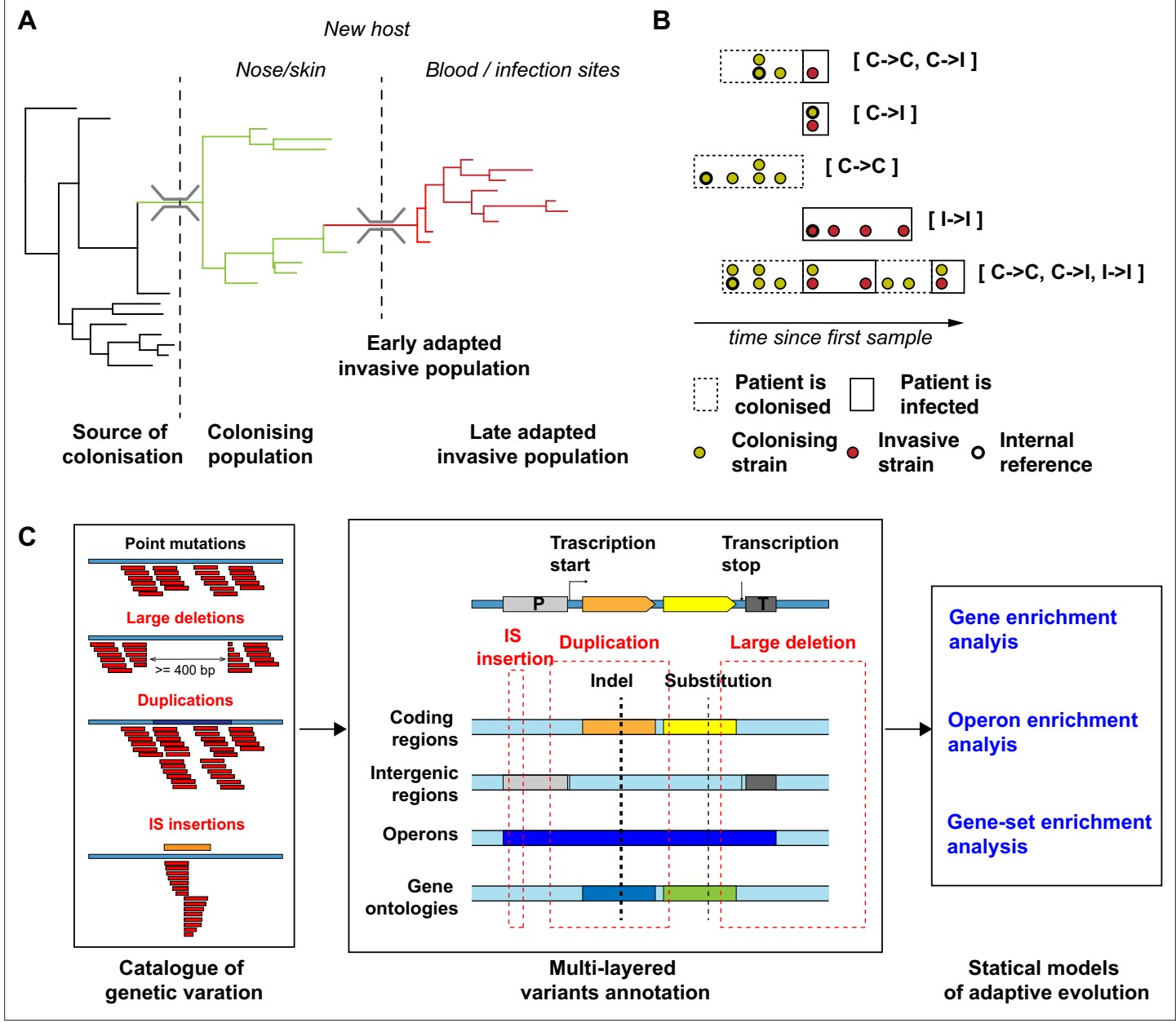

**Figure 1.** Overview of the *S. aureus* within-host evolution analysis framework. (**A**) Simulated phylogenetic tree illustrating within-host evolution of *S. aureus* colonisation and infection. This model assumes two genetic bottlenecks (dotted lines); upon transmission and upon transition from colonisation to invasive infection. (**B**) Sites and timing of within-host samples and number of genomes per sample define five prototypes of within-host evolution studies, each with colonising-colonising (C>C), colonising-invasive (C>I), or invasive-invasive (I>I) comparisons in different combinations: from top to the bottom: multiple colonising samples and one invasive samples; one colonising and one invasive sample; multiple colonising samples; multiple invasive samples; multiple colonising and invasive samples. (**C**) Approach to capture signals of adaptation across multiple independent episodes of colonisation/infection through detection of multiple genetic mechanisms of adaptation from short reads data and multi-layered functional annotation of the genetic variants using multiple databases including characterisation of intergenic regions (promoters), operon prediction, and gene ontology (GO). Statistical framework for the gene, operon, and gene set enrichment anlaysis (GSEA). Counts of independent mutations with likely impact on the protein sequence (non-synonymous substitutions, frameshifts, stop codon mutations, and insertion sequences [IS] insertions) were computed for each genes with a FPR3757 homologue. Gene counts (with the addition of intergenic mutations in promoter regions) were aggregated in operons and GOs. Gene and operon counts were used to fit Poisson regression models to infer mutation enrichment and significance of the enrichment. GOs counts and gene enrichment significance were used to run a gene-set-enrichment analysis. To illustrate the approach, the example of the gene *walR* is provided in italic.

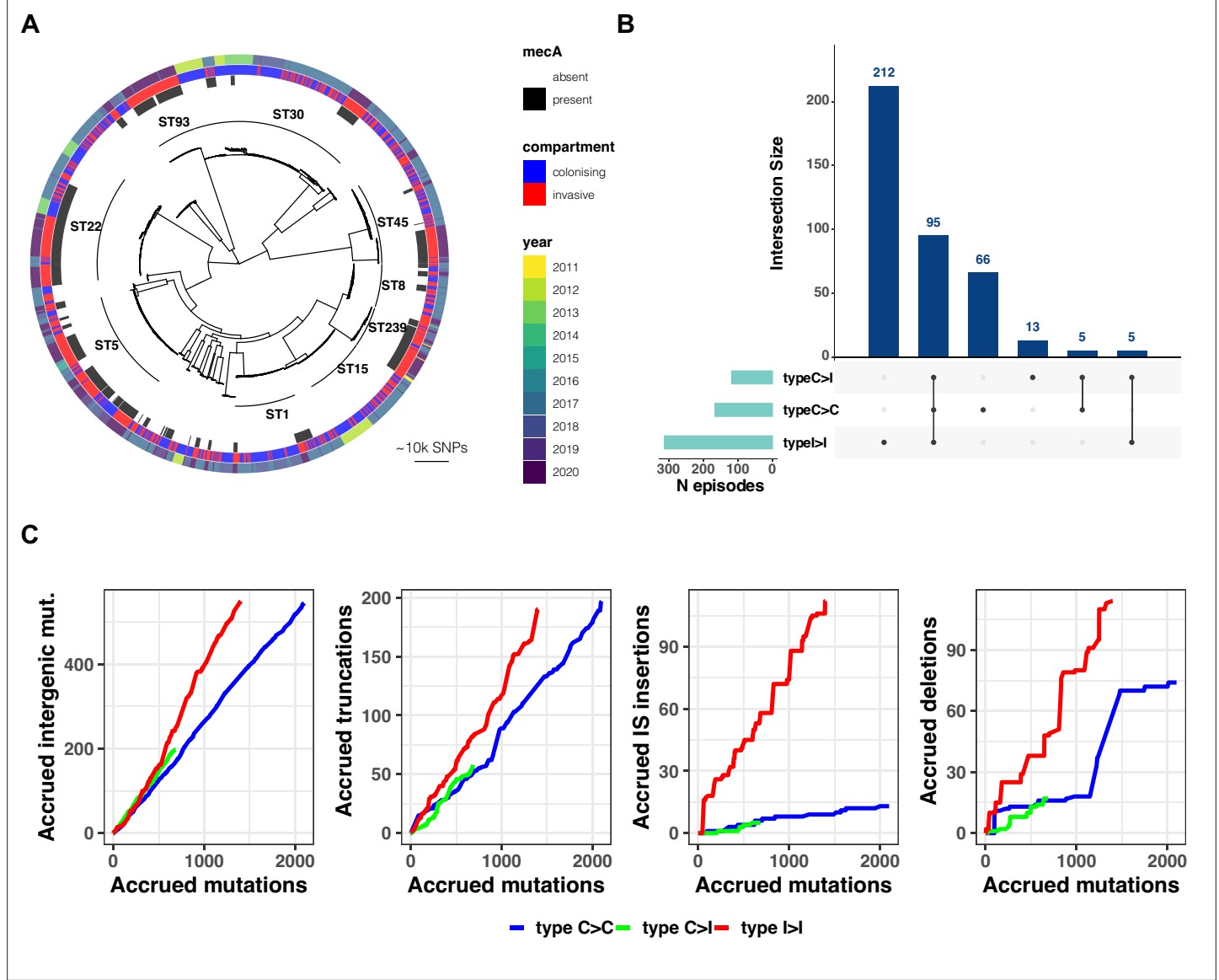

**Figure 2.** (**A**) Maximum-likelihood phylogenetic tree of 2590 *S. aureus* sequences included in the study. The tree is annotated (starting from the inner circle) with the most prevalent sequence types (ST), presence/absence of the *mecA* gene, compartment of isolation (colonising or invasive), and year of publication. (**B**) Summary of 396 independent episodes of *S. aureus* colonisation or infection categorised according to whether they allowed comparing colonising-colonising (C>C), colonising-invasive (C>I), or invasive-invasive (I>I) strains, or a combination of them. (**C**) Evidence of a distinctive pattern of adaptation in late infection-adapted strains (type I>I variants). For each type of comparison (type C>C, colonising-colonising; type C>I, colonising-invasive; type I>I, invasive-invasive), the cumulative curves display the accrued number of intergenic mutations, truncating mutations, insertion sequences (IS) insertions, and large deletions as a function of the total number of mutations. Genetic events were counted once per episode, regardless of the number of strains with the mutation. The sequence of mutations events in the cumulative curves is random.

The online version of this article includes the following figure supplement(s) for figure 2:

**Figure supplement 1.** Number of episode-specific variants in same-episode strains having the same sequence type (ST) as the internal reference vs. isolates with a different ST.

**Figure supplement 2.** Correlation between number of samples per episode and mean mutation counts.

**Figure supplement 3.** Within-host mutation rates within the colonising and invasive populations.

**Figure supplement 4.** Regression diagnostics to assess linear regressions sampling time after the internal reference and number of mutations.

**Figure supplement 5.** Distribution of new IS insertions by classification of the transposase and by major sequence types (ST).

312 type I>I within-host variants. In 95 cases, there were sufficient samples to assess all three types within the same episode (*Figure 2B*). Across colonisation/infection stages, sampling frequency did not seem to affect the number of variants identified with the exception of early adapted invasive strains (*Figure 2—figure supplement 2*).

We first sought to explore whether the rate of mutations over time differs between colonising and invasive populations. To estimate within-host mutation rates, we fitted linear regressions using data from a subset of 109 episodes that had at least two colonising or invasive isolates collected at two different timepoints (total 701 strains). The regressions suggested that mutation rates were higher in the invasive population; however, this analysis was limited by the heterogeneity of sampling strategies and by evidence of heteroskedasticity and non-linear distribution (*Figure 2—figure supplements 3 and 4*).

We have previously shown that invasive strains from persistent or relapsing infections exhibit a high proportion of protein-truncating mutations (*Giulieri et al., 2018*). A similar enrichment of protein-truncating variants was identified within invasive strains as compared to strains from asymptomatically colonised individuals (*Young et al., 2017*; *Young et al., 2012*). We reasoned that if this indicates genome degradation during infection, infecting strains might also be enriched for other loss of function (LOF) mutations caused by structural variants, such as movement of IS (*Hawkey et al., 2020*) and large deletions, leading to complete or partial gene loss (*Toft and Andersson, 2010*). In addition, we hypothesised that mutations and IS insertions in intergenic regions might contribute to altering gene expression or activity by interfering with the expression of key genes or operons (*McEvoy et al., 2013*).

Therefore, we calculated the prevalence of intergenic mutations, protein-truncating mutations, IS insertions, and large deletions among all variants and compared it between type C>C, type C>I, and type I>I variants. Strikingly, the distribution of mutations according to the predicted effect differed substantially in I>I pairs when compared to mutations identified between nose-colonising and invasive strains and within colonising strains (*Figure 2*). This can be expressed using the neutrality index (NI), which tests deviation from neutral evolution and is comparable to an odds ratio (*Stoletzki and Eyre-Walker, 2011*). Relative to type C>C variants, variants emerging within the infecting strains were enriched for intergenic mutations (NI 2.5; $p=1.8 \times 10^{-16}$) and protein-truncating mutations (NI 2.4; $p=4.8 \times 10^{-10}$) (*Table 2*). In contrast, no significant enrichment was observed among type C>I variants.

While large deletions were significantly more enriched in type I>I variants (NI 4.0, $p=1.1 \times 10^{-15}$), the strongest evidence for enrichment (NI 19.9, $p=1.6 \times 10^{-42}$) was found for IS insertions. We and others have previously shown that new insertions of IS*256* may provide an efficient mechanism of genomic plasticity in invasive *S. aureus* strains (*Giulieri et al., 2018*; *Kuroda et al., 2019*; *McEvoy et al., 2013*). Here, we expand this observation in a larger dataset and show that this mechanism is not limited to IS*256* (*Figure 2—figure supplement 5*). As shown in *Figure 2C*, two invasive strains exhibited a burst of >10 new IS insertions (IS*3* and IS*256*, respectively). It has been shown that IS activation occurs under stress conditions, such as antibiotic exposure and oxidative stress (*Schreiber et al.,*

**Table 2.** Modified McDonald-Kreitman table displaying counts of variants (point mutations and structural variants) and the neutrality index for colonising-invasive (type C>I) and invasive-invasive (type I>I) variants (both compared to colonising-colonising [type C>C] variants).

| Classification of variant | Number of variants *(Neutrality index)* | | |
| --- | --- | --- | --- |
| | Type C>C | Type C>I | Type I>I |
| Synonymous | 381 | 130 | 155 |
| Non-synonymous | 978 | 300 *(0.9)* | 503 *(1.3)*\* |
| Intergenic | 544 | 197 *(1.1)* | 549 *(2.5)*\*\* |
| Truncating | 197 | 58 *(0.9)* | 190 *(2.4)*\*\* |
| Insertion sequences insertion | 17 | 6 *(1.0)* | 137 *(19.8)*\*\* |
| Large deletion | 76 | 17 *(0.6)*\* | 122 *(3.9)*\*\* |

Values are counts of independent mutations. The neutrality index is shown in brackets in italic.
Significance testing Fisher's Exact Test: $p<0.05$; \*\* $p<0.005$.

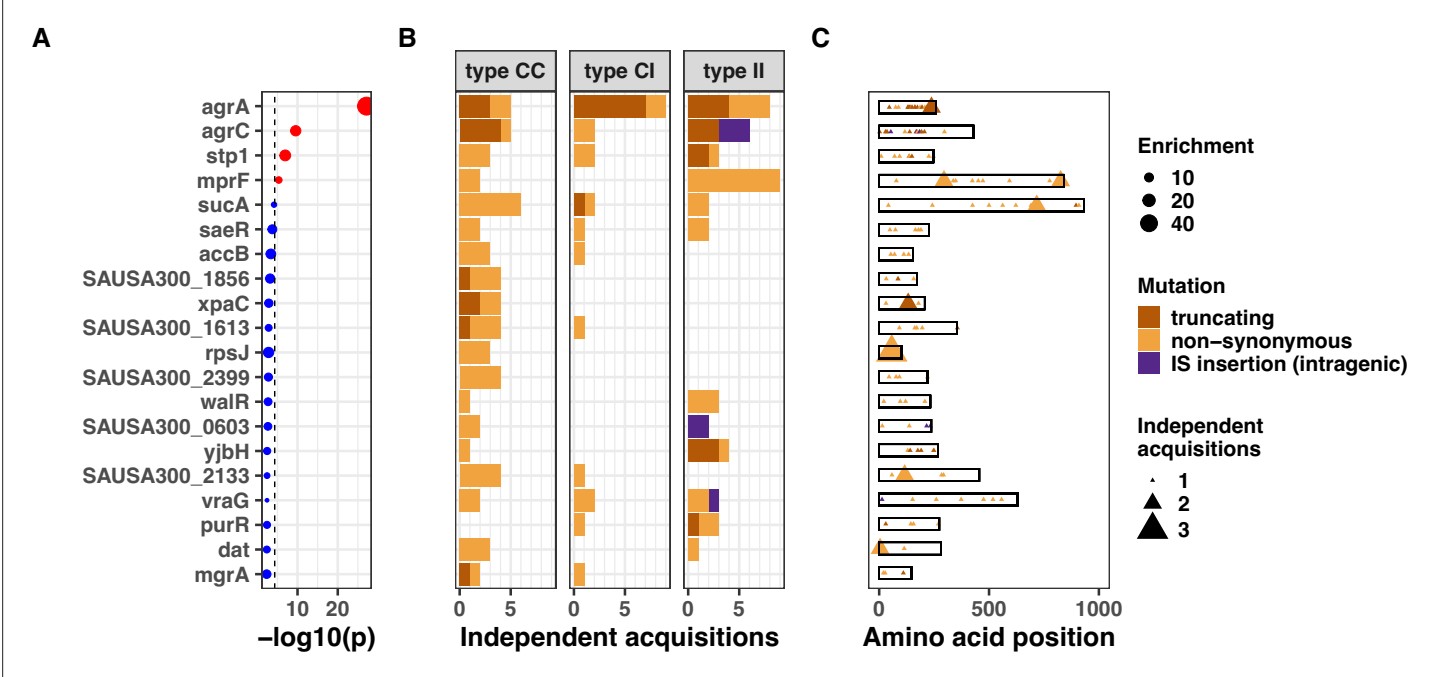

**Figure 3.** Top 20 genes with the most significant mutation enrichment across the entire dataset. (**A**) Significance of the enrichment for protein-altering mutations. The dashed line depicts the Bonferroni-corrected significance threshold, and red circles and blue circles represent genes with p values below and above the Bonferroni threshold, respectively. (B) Bar plots of independent mutations separated in three panels according to the type of variant (type C>C: colonising-colonising; type C>I: colonising-invasive; type I>I: invasive-invasive) and coloured according to the class of mutation. (C) Gene maps with type and positions of mutations.

The online version of this article includes the following figure supplement(s) for figure 3:

**Figure supplement 1.** Mapping of mutations in the 10 most significantly enriched mutated genes across the entire dataset.

**Figure supplement 2.** dN/dS values for non-synonymous mutations (**A**), indels (**B**), and non-sense mutations (stop codons) (**B**) for FPR3757 genes.

**Figure supplement 3.** Scatter plot representing in silico inferred functional impact of variants in the 20 most convergent loci.

**Figure supplement 4.** Most frequently deleted genes in large deletions.

**Figure supplement 5.** Most frequently enriched genes in copy number variations.

**Figure supplement 6.** Gene convergence analysis of all mutated genes (i.e. including both genes with FPR3757 homologue and no FPR3757 homologue).

**Figure supplement 7.** Gene convergence analysis after removing variants in strains included in *Young et al., 2017*, the largest collection of this analysis (1078 strains and 105 episodes).

*2013*), which is consistent with the selection environment encountered by invasive strains. However, these bursts occurred only in 2/1068 adapted invasive strains.

Overall, these data support a model, where late infection-adapted strains show an enrichment for variants that are predicted to exert a stronger functional impact, either by producing a truncated protein or by potentially interfering with intergenic regulatory regions, through point mutations or IS insertions. This strong genome degradation signature appears to be specific to type I>I variants and was absent in type C>I variants, suggesting that the bottleneck effect upon blood or tissue invasion does not explain it. To assess whether this general enrichment of non-silent evolution represented a signature of positive selection or derived from within-host gene obsolescence occurring during invasive infection, we further investigated signals of gene, operon, and pathway specific enrichment across independent episodes of infection.

## Gene enrichment analysis identifies significant hotspots of adaptation

To identify signatures of adaptation, we first counted how many times each coding sequence was mutated independently across distinct colonisation/infection episodes (*Figure 3*, *Table 3*, *Table 4*, *Supplementary file 3*). We considered all protein-modifying mutations either predicted to cause a

**Table 3.** Genome-wide significant gene signatures of within-host evolution.

The genes shown reached genome-wide significance in the entire dataset or in either colonising-colonising (type C>C), colonising-invasive (type C>I), or invasive-invasive (type I>I) variants.

| Gene | p value (whole dataset) | Description | N independent mutations | | | Significance |
| | | | Type C>C | Type C>I | Type I>I | |
|---|---|---|---|---|---|---|
| agrA* | $7.04 \times 10^{-28}$ | Accessory gene regulator protein A | 5** | 9** | 8** | Part of the agr quorum sensing system, which is the master regulator of virulence factors expression in *S. aureus*. Recurrent mutations associated with invasive disease. |
| agrC** | $2.84 \times 10^{-10}$ | Accessory gene regulator protein C | 4 | 2 | 6** | Histidine kinase, receptor for extracellular autoactivating peptide. Phosphorylates agrA. |
| stp1** | $1.13 \times 10^{-7}$ | Protein phosphatase 2 C domain-containing protein | 3 | 2 | 3 | Associated with vancomycin resistance. |
| mprF** | $4.55 \times 10^{-6}$ | Oxacillin resistance-related FmtC protein | 2 | 0 | 9** | Main determinant of daptomycin resistance. Association with persistence and immune evasion. |
| rpoB | $7.24 \times 10^{-3}$ | DNA-directed RNA polymerase subunit beta | 1 | 1 | 7** | Association with rifampicin resistance, but selection in the absence of rifampicin exposure can happen (R503H). Co-resistance to vancomycin, daptomycin, and oxacillin. Association with persistence. |

*Significant enrichment (above the Bonferroni-corrected cut-off, see methods).

gain or LOF to the locus: non-synonymous substitutions, truncations, IS insertions, or deletions. To ensure consistency across the dataset, we restricted our analysis to 1736 (74%) genes with a homologue in reference strain FPR3757 (excluding plasmid genes and phage genes). Mutations were considered independent if they arose in distinct colonisation/infection episodes. To assess whether the convergent signals were a reliable indication of adaptation, we applied a gene enrichment analysis for protein-altering mutations which computes a length-corrected gene-level enrichment of protein-modifying mutations. The significance of the enrichment for each gene was estimated by comparing gene-specific Poisson models of mutation counts with the null hypothesis, which indicates neutrality and assumes a constant mutation rate across all genes (*Young et al., 2017*).

When applying a Bonferroni-corrected significance threshold ($4.6 \times 10^{-5}$), mutations in *agrA* were highly significantly enriched across the entire dataset (45-fold enrichment, p=$7.0 \times 10^{-28}$). Other significantly enriched genes were *agrC* (13-fold enrichment, p=$2.8 \times 10^{-10}$), *stp1* (14-fold enrichment, p=$1.1 \times 10^{-7}$), and *mprF* (sixfold enrichment, p=$4.6 \times 10^{-6}$). The gene *sucA* reached near-significance (fivefold enrichment, p=$6.8 \times 10^{-5}$). Mutations in genes most significantly targeted by convergent evolution were evenly distributed across the *S. aureus* phylogeny, indicating that these adaptative mechanisms were not specific to selected lineages (*Figure 3—figure supplement 1*). Using dN/dS analysis, we confirmed signatures of positive selection in the most significantly enriched genes, although only *agrA* reached statistical significance (*Figure 3—figure supplement 2* and *Supplementary file 4*).

We found that several genes with the most significant enrichment (*agrA*, *agrC*, *stp1*, and *sucA*) were recurrently mutated across all three within-host evolutionary scenarios, implying a global role in *S. aureus* adaptation during colonisation and invasion. This suggests partial adaptation of *S. aureus* strains upon invasion. It has been previously shown that adaptive mutations, particularly within the quorum sensing accessory gene regulator (*agr*), are enriched in invasive *S. aureus* strains, suggesting that adapted strains are more prone to be involved in invasive disease (*Young et al., 2017*; *Young et al., 2012*; *Altman et al., 2018*; *Smyth et al., 2012*). While the key role of *agr* was consistent with previous evidence from clinical and experimental studies, the high number of recurrent *sucA* mutations was surprising. This metabolic gene encodes for the α-ketoglutarate dehydrogenase of the tricarboxylic acid cycle, and recent work has revealed the functional basis of its potential role in adaptation. Its inactivation was found to lead to a persister phenotype (*Wang et al., 2018*), and *sucA* was a hotspot of metabolic adaptation to antibiotics in a recent in vitro evolution study (*Lopatkin et al., 2021*).

Adaptive mutations can cause both loss or gain of function of the gene affected; however, LOF is thought to be most frequent consequence of adaptation (*Behe, 2010*). We inferred LOF when variants lead to protein truncation (including intragenic IS insertions) or non-synonymous substitutions were expected to be deleterious based on the degree of divergence in conserved sites of the protein

**Table 4.** Gene signatures of within-host evolution with suggestive significant enrichment.

The genes shown reached the suggestive significance threshold in the entire dataset or in either type C>C, type C>I, or type I>I variants.

| Gene | p value (whole dataset) | Description | N independent mutations | | | Significance |
|------|------|------|------|------|------|------|
| | | | Type C>C | Type C>I | Type I>I | |
| sucA* | $6.82 \times 10^{-5}$ | 2-oxoglutarate dehydrogenase E1 component | 6 | 2 | 2 | Encodes a subunit of the α-ketoglutarate dehydrogenase of the tricarboxylic acid cycle. |
| saeR* | $1.83 \times 10^{-4}$ | DNA-binding response regulator SaeR | 2 | 1 | 2 | Regulator component of the saeRS two-component system. Virulence regulation. |
| accB | $4.27 \times 10^{-4}$ | Biotin carboxyl carrier protein of acetyl-CoA carboxylase | 3* | 1 | 0 | Part of the fatty acid synthesis pathway of S. aureus. |
| SAUSA300_1856 | $6.41 \times 10^{-4}$ | Hypothetical protein | 4* | 0 | 0 | Intracellular cysteine peptidase. Putative chaperone in S. aureus. |
| xpaC | $1.38 \times 10^{-3}$ | Hypothetical protein | 4* | 0 | 0 | Predicted 5-bromo-4-chloroindolyl phosphate hydrolysis protein, no data on S. aureus. |
| rpsJ | $1.58 \times 10^{-3}$ | 30S ribosomal protein S10 | 3* | 0 | 0 | Mutations at residues 53–60 are associated with tigecycline resistance, at no apparent fitness cost. |
| SAUSA300_2399 | $1.68 \times 10^{-3}$ | ABC transporter ATP-binding protein | 4* | 0 | 0 | Downregulated in the presence of fusidic acid |
| walR | $2.10 \times 10^{-3}$ | DNA-binding response regulator | 1 | 0 | 3* | Part of walKR two-component response regulator. Associated with vancomycin resistance. |
| yjbH | $3.55 \times 10^{-3}$ | Dsba-family protein | 1 | 0 | 3* | Negative regulator of spx (directs its ClpXP-dependent degradation). Association with antibiotic resistance, virulence regulation, and oxidative stress resistance. |
| purR | $3.86 \times 10^{-3}$ | Pur operon repressor | 0 | 1 | 3* | purR mutants: increased biofilm formation and virulence in animal model; higher capacity to invade epithelial cells. |
| era | $5.34 \times 10^{-3}$ | GTP-binding protein Era | 0 | 1 | 3* | Involved in ribosome assembly and stringent response. |
| pbp2 | $7.75 \times 10^{-3}$ | Penicillin-binding protein 2 | 6* | 0 | 0 | Role in methicillin resistance (PBP2a synergism). Increased expression after oxacillin exposure. |
| fakA | $9.90 \times 10^{-3}$ | Hypothetical protein | 5* | 0 | 0 | Fatty acid kinase. Deletion mutant displayed increased virulence in a murine model of skin infection. |
| sgtB | $2.65 \times 10^{-2}$ | Glycosyltransferase | 0 | 0 | 3* | sgtB mutations in adaptive laboratory evolution experiments upon vancomycin exposure. |

*suggestive significant enrichment (above the suggestive significance cut-off, adjusted for false-discovery, see methods).

**Choi et al., 2012**. The majority (63%) of aggregated variants in adaptive loci were predicted to lead to LOF (**Figure 3—figure supplement 3**), either because of accumulation of truncating mutations (agrA and agrC) or exclusively because of deleterious substitutions (walR and vraG). Because only LOF is inferred from the sequence, gain of function variants might be miss-classified as neutral (or even deleterious). We considered this hypothesis for genes with low frequency of truncations and expected adaptive advantage of gain of function mutations based on the literature. For example, mprF mutations associated with vancomycin or daptomycin resistance have been shown to be associated with increased enzymatic activity of the protein leading to decreased negative charge of the membrane (**Ernst and Peschel, 2019**). Among within-host acquired mprF mutations in our dataset, we found that four of them were previously described and associated with antibiotic resistance through a gain of function mechanism (**Ernst and Peschel, 2019**). A distinctive sign of these variants was convergence at position or mutation level, which has been described as a potential hallmark of gain of function mutations (**Gerasimavicius et al., 2021**). Further supporting the evidence for gain of function, mprF was duplicated in one of the episodes, as reported previously (**Gao et al., 2015**), while no other convergent gene was affected by within-host copy number variants. We also assessed accB because no variant in this genes was classified as deleterious; however, accB mutations have been previously described as LOF in strains that are auxotroph for fatty acids (**Morvan et al., 2016**). Thus, based on in silico prediction and previous data, we hypothesise that variants in convergent loci were mostly expected to be LOF with the exception of mprF.

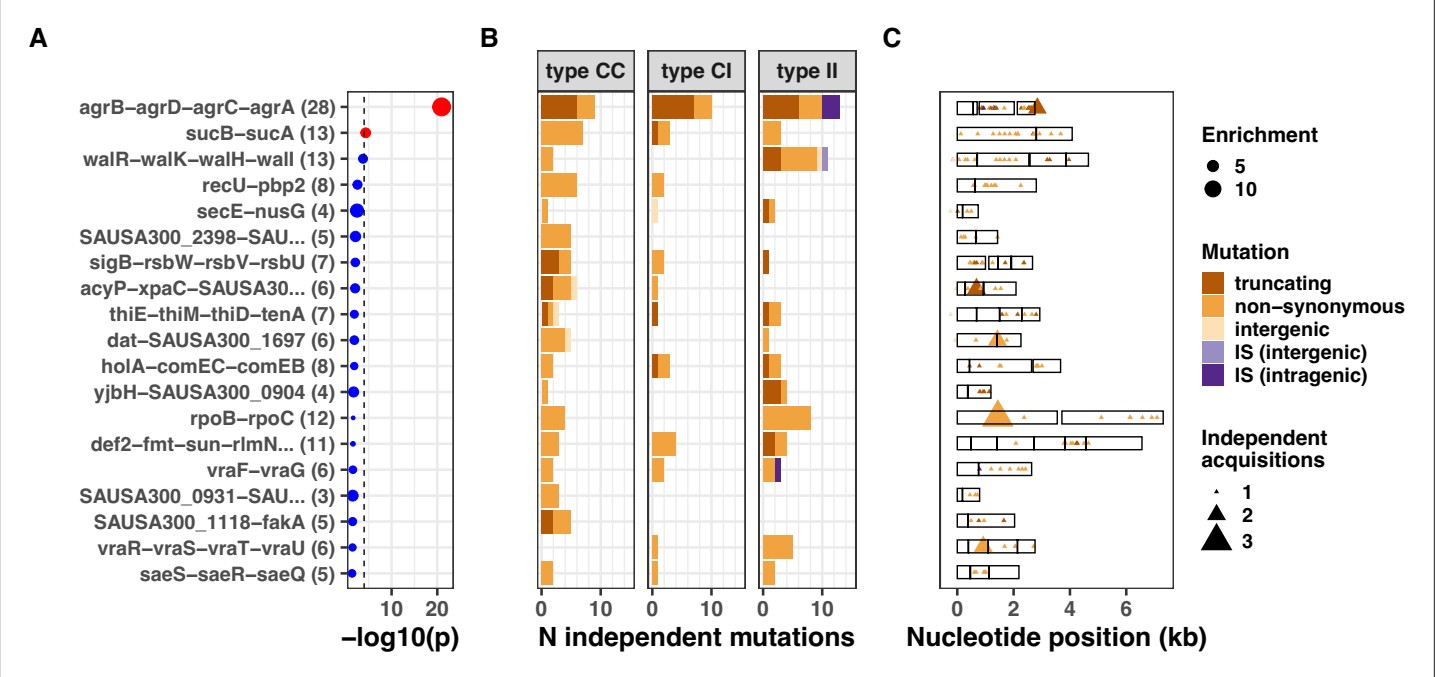

**Figure 4.** Top 20 operons with the most significant mutation enrichment across all dataset. (**A**) Significance of the enrichment for protein-altering mutations. The dashed line depicts the Bonferroni-corrected significance threshold, and red circles and blue circles represent operons with p values below and above the Bonferroni threshold, respectively. (**B**) Bar plots of independent mutations separated in three panels according to the type of variant (type C>C: colonising-colonising; type C>I: colonising-invasive; type I>I: invasive-invasive) and coloured according to the class of mutation. Mutations were considered independent if they occurred in separate episodes of either colonisation or invasive infection. (**C**) Operon maps with positions of the mutations (relative to the start of the first gene of the operon). Operons are labelled with the names of the genes included, and longer labels were shorted for clarity (see **Supplementary file 5** for details).

To confirm that our gene enrichment analysis (focused on point mutations and IS insertions and limited to genes with FPR3757 homologues) captured the large part of adaptation, we analysed variation due to large deletions and copy number variation, which were not included in the gene enrichment analysis. We observed multiple independent deletions and amplifications mainly in phage genes (**Figure 3—figure supplement 4** and **Figure 3—figure supplement 5**). We also repeated the gene enrichment analysis with all mutated genes (with and without FPR3757 homologues) and found very similar results, with only two hypothetical proteins with no FPR3757 homologue among the genes with most significant enrichment (**Figure 3—figure supplement 6**).

## Combining multiple mechanisms of adaptation and multi-layered annotation increases the signal of convergent evolution

To increase our ability to capture signatures of adaptation from convergent evolution, we extended our analysis beyond coding sequences, to integrate the genetic variation signals issued from intergenic mutations and IS insertions in intergenic regions. This multi-layered annotation of mutated regions was shown to increase the amount of information gained from in vitro adaptive evolution experiments (**Phaneuf et al., 2020**). Such methodology allows for an advanced classification of intergenic mutations based on regulatory sequences including promoters and transcription units based on data acquired from RNAseq experiments (**Mäder et al., 2016**; **Prados et al., 2016**).

Using this approach, we were able to assign 150/1237 (11%) of all intergenic mutations and IS insertions to a predicted regulatory region. We found that the *agr*, *sucAB*, and *walKR* operons had the strongest convergent evolution signal with 28, 13, and 13 independent mutations (**Supplementary file 5**). Mutations within these loci were significantly (*agr*, 12-fold enrichment, p=1.1e-21; *sucAB*, fourfold enrichment, p=4.5e-5) or near-significantly enriched (*walKR*, threefold enrichment, p=1.7e-4) (**Figure 4**). Interestingly, promoter mutations represented 2/13 (15%) of the *walKR* operon mutations, indicating that potentially impactful intergenic variants may be missed when considering only

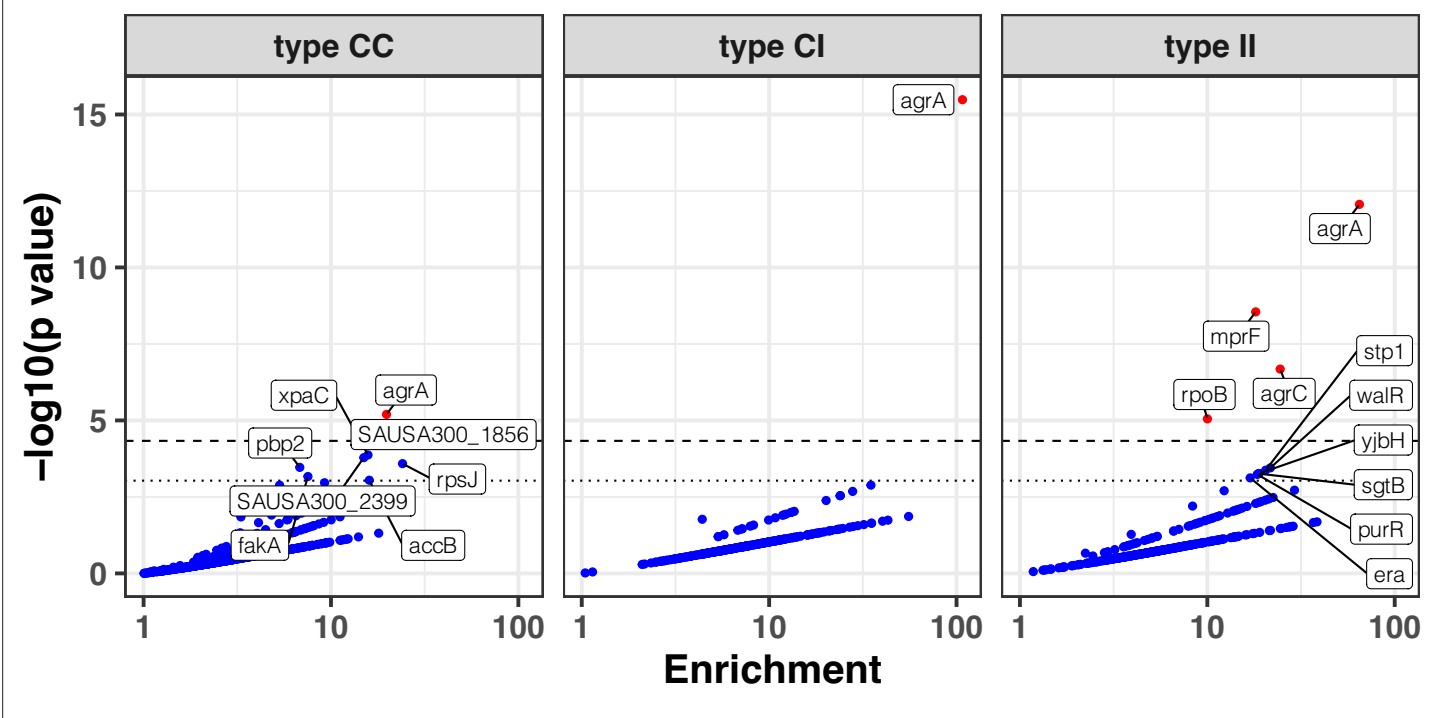

**Figure 5.** Modified volcano plot displaying enrichment (x-axis) and significance of enrichment (y-axis) within colonising-colonising (type C>C), colonising-invasive (type C>I), and invasive-invasive (type I>I) variants. The horizontal dashed line depicts the Bonferroni-corrected significance threshold and dotted line shows the suggestive significance threshold. Labels indicate genes with significance of enrichment below the suggestive threshold. Genes are coloured in red if the p value is below the Bonferroni-corrected threshold and in blue otherwise.

The online version of this article includes the following figure supplement(s) for figure 5:

**Figure supplement 1.** Modified volcano plot displaying enrichment (x-axis) and significance of enrichment (y-axis) for FPR3757 operons across the entire dataset, colonising-colonising (type CC), colonising-invasive (type CI), and invasive-invasive (type II) variants.

**Figure supplement 2.** Gene set enrichment analysis (GSEA) for protein-modifying mutations in colonising-colonising (type CC), colonising-invasive (type CI), and invasive-invasive (type II) variants.

coding regions. Furthermore, new IS insertions were found within type I>I variants: three insertions into *agrC* (predicted to inactivate the gene, as shown previously in staphylococci [***Both et al., 2021***; ***Suligoy et al., 2020***]) and an insertion 159 bp upstream of *walR*, in a region encompassing its cognate promoter. Together with the strong enrichment for IS insertions within type I>I variants, the location of these insertions in recurrently mutated operons suggests that IS insertions contribute to the adaptive evolution of *S. aureus* during invasive infection.

## Adaptation within the invasive population is distinctive and strongly driven by antibiotics

The excess of non-silent evolution (and potentially function-altering) within invasive strains suggested that strong, specific selection pressure occurs within the invasive populations (type I>I variants). We therefore assessed genes that appeared to be specifically mutated or inactivated during infection. We performed our gene- and operon-enrichment analysis for each type of within-host variants separately (i.e. within the colonising population, between colonising and invasive strains, and within the invasive population) (***Figure 5***). We found that *agrA* mutations were highly enriched in any group of variants, and particularly prevalent between colonising and invasive strains (type C>I variants), consistent with a previous study that is included in this analysis (***Young et al., 2017***). Among type I>I variants (between invasive strains), a significant enrichment was observed in *mprF* (18-fold enrichment, p=2.8 × 10⁻⁹), *agrC* (24-fold enrichment, p=2.1 × 10⁻⁷), and *rpoB* (10-fold enrichment, p=8.8 × 10⁻⁶). Other genes that were strongly enriched in type I>I variants (below the Bonferroni-corrected threshold, but above the suggestive significance threshold, ***Figure 5***) included *walR* (22-fold enrichment, p=3.5 × 10⁻⁴), *stp1*

(20-fold enrichment, p=4.2 × 10⁻⁴), *yjbH* (19-fold enrichment, p=5.4 × 10⁻⁴), *sgtB* (19-fold enrichment, p=5.5 × 10⁻⁴), and *purR* (18-fold enrichment, p=5.8 × 10⁻⁴).

The enrichment for mutations in *mprF, rpoB, stp1, sgtB,* and in the *walKR/yycH* operon (11-fold enrichment, p=9 × 10⁻⁹, see *Figure 5—figure supplement 1* for the operon enrichment analysis) highlights the role of antibiotic pressure in shaping adaptation within the invasive population, since these loci are hotspots of adaptation to key anti-staphylococcal antibiotics that are often used in invasive infections (rifampicin, daptomycin, and vancomycin). For example, the essential two-component regulator *walKR/yycFG* (and its associated genes *walH/yycH*) have been shown to have a key role in vancomycin resistance in one of the within-host evolution studies included in this analysis (*Howden et al., 2011*), while mutations in both *stp1* and *sgtB* have been observed in vancomycin-adapted strains (*Machado et al., 2021*).

Notably, the most significant gene signatures in invasive strains might have been selected in response to other selective pressures, including the host immune response during infection. For example, *rpoB* mutations have been associated with pleiotropic effects, including co-resistance to vancomycin, daptomycin, and oxacillin and immune evasion, suggesting a potential role in adaptation beyond the response to the selective pressure from rifampicin (*Guérillot et al., 2018*). This hypothesis is supported by the presence of mutations (such as the *rpoB* R503H substitution and N405 inframe deletion) outside the rifampicin-resistance determining region.

Pleiotropic phenotypes are also likely to underlie the enrichment of *yjbH* with invasive strains, which was mutated four times (of which three were truncations), yet only one mutation was found in colonising strains or early infection-adapted strains. This gene has a cysteine-rich domain that is homologous to *dsbA* in *Escherichia coli*. One of its roles in *S. aureus* is to facilitate the ClpXP-dependent degradation of the transcriptional regulator Spx (*Austin et al., 2019*). Inactivation of *yjbH* has been associated with oxacillin (*Göhring et al., 2011*) and vancomyin (*Renzoni et al., 2011*) resistance, impaired growth (*Engman et al., 2012*), and reduced virulence in animal models (*Paudel et al., 2021*), indicating that *yjbH* mutations may influence both host-pathogen interaction and antibiotic resistance. Finally, *purR*, a purine biosynthesis repressor, has been recently characterised beyond its metabolic function: interestingly, it was shown to be a virulence regulator (*Sause et al., 2019*), where *purR* mutants displayed higher bacterial counts following mice infection, increased biofilm formation (*Goncheva et al., 2019*), and higher capacity to invade epithelial cells (*Goncheva et al., 2020*).

We performed a gene set enrichment analysis (GSEA), using gene ontology and antibiotic resistance gene annotations (*Feldgarden et al., 2019*). The GSEA, stratified by variant type, showed significant enrichment only in type I>I variants, further underscoring the higher level of adaptation in this group (*Figure 5—figure supplement 2* and *Supplementary file 6*) and confirmed the broad functional implications of the most enriched genes and operons with the invasive populations, since among the ontologies that were significantly enriched within the invasive population, we found the categories 'DNA binding' (normalised enrichment score [NES]=1.6, false discovery rate [FDR]-adjusted p=9 × 10⁻⁴), 'pathogenesis' (NES = 1.7, adjusted p=4 × 10⁻³), and 'antibiotic response' (NES = 1.8, adjusted p=7 × 10⁻³).

Taken together, these findings point to six key genetic loci that appear to have an important role in *S. aureus* adaptation during invasive infections. These loci are associated with either antibiotic resistance (*mprF, rpoB, stp1, sgtB,* and *walKR*), pathogenesis (*agrAC* and *purR*), or both (*yjbH*).

## A mutation's co-occurrence network defines loci under within-host co-evolutionary pressure

Epistasis, defined as the interaction of multiple mutations on a given phenotype (*Levin-Reisman et al., 2019*), plays a role in adaptive evolution in bacteria, particularly in antibiotic resistance (*Skwark et al., 2017*; *Wadsworth et al., 2018*; *Yokoyama et al., 2018*). Whether epistatic interactions could promote *S. aureus* adaptation during infection remains unknown. Identifying these interactions would enable identification of combinations of mutations underlying bacterial adaption during infection and refine the prediction of infection outcomes. Here, we assessed co-occurrence of mutations and mutated genes across independent episodes of colonisation/infection. While co-occurrence may simply result from co-selection (e.g. simultaneous exposure to two different antibiotics), it may also indicate putative epistatic interactions that could be explored in terms of potential impact on adaptive phenotypes (*Phillips, 2008*).

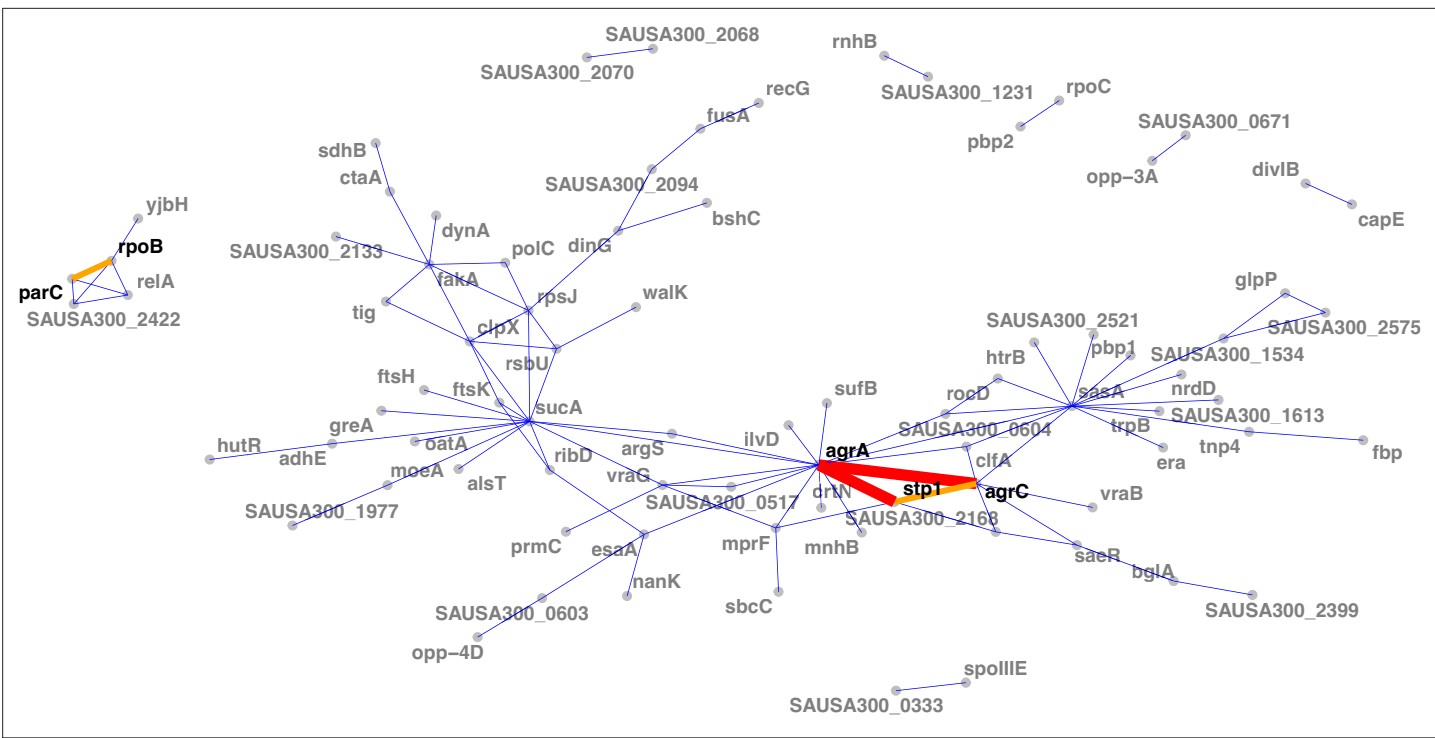

**Figure 6.** Network of mutations co-occurrence. The width and colour of the edges represent the strength of the co-occurrence of mutated genes on the same strain (thin and blue, two independent co-occurrences; thick and orange, three independent co-occurrences).

First, we explored co-occurrence of mutations and found only one case where the same mutations co-occurred in more than one independent episode. The two mutations were an inframe deletion within hypothetical protein SAUSA300_2068 and a A60D substitution of the gene *ywlC*. These genes are closely located in FPR3757. While the co-occurrence could be explained by recombination, recombination is expected to be rare amongst within-host *S. aureus* populations in general (*Golubchik et al., 2013*) and even rarer within invasive strains. YwlC is a threonylcarbamoyl-AMP synthase *in E. coli*, thus it is possible that SAUSA300_2068 is also a ribosomal protein. Ribosomal proteins can display regulatory activity (*Aseev and Boni, 2011*) and could plausibly be targets of adaptation to both antibiotics and the host/intracellular survival. This specific case of convergent co-occurrence of mutations was detected within type I>I variants.

When assessing interactions at gene level (i.e. co-occurrence of the same altered protein sequences across independent episodes), we found the strongest interaction between the *agrA* and *agrC* genes (*Figure 6*). While this is consistent with the high convergence of mutations in the *agr* locus, this suggests that strains acquire multiple mutations within the locus, possibly further impacting *agr* activity. Interestingly, no convergent co-occurrence signature compatible with possible epistasis was observed within the *walKR* locus, the other operon with a high number of independent mutations; which could be due to the essentiality of *walKR* in *S. aureus* (*Monk et al., 2019*). Collectively, *agr* locus mutations interacted with 17 other mutated genes, the strongest interaction being with *stp1*. Since *stp1* (a serine/threonine phosphatase) has been previously associated with virulence regulation (*Cameron et al., 2012*), this interaction potentially indicates another mechanism by which adapted strains fine-tune the gene expression profile that is already altered by *agr* mutations.

Another moderately strong interaction was observed between *rpoB* and *parC*, which were co-mutated in three independent episodes. Given the association of *parC* mutations with fluoroquinolone resistance (*Trong et al., 2005*), this interaction is likely to be an example of co-selection due to co-exposure to fluoroquinolones and rifampicin.

## Clinical correlates of adaptive signatures within colonising and invasive populations

Genetic signatures of bacterial adaptation have been associated with infection extent, for example, enabling the prediction of extraintestinal infection with *S. enterica* (*Wheeler et al., 2018*). We have previously shown that adaptive mutations are enriched in invasive infections; however, it is unclear whether bacterial adaptation is more likely to be associated with distinctive clinical syndromes. To identify clinical correlates of adaptive signatures, we classified colonisation and infection episodes based on the sites of collection and on clinical data obtained from the publications (*Table 1* and *Figure 7—figure supplement 1*). We then used the Jaccard index and network analysis to compute node centrality as a global measure of adaptation for each independent episode. The Jaccard index can be used as a simple marker of the proportion of shared mutated genes between pairs of colonisation or infection episodes (*Bailey et al., 2017*). Node centrality allows to similtuaneously take into account the strength of similarity between independent episodes (Jaccard index) and the number of pairs with shared mutated genes (number of connections). Hence, a node centrality of 0, indicates that the episode does not share any mutated genes with other episodes and appears as isolates node on the adaptation network (*Figure 7—figure supplement 2*). Here, we limited the analysis to the 20 most significantly enriched genes with each type of variant.

Our network analysis showed that adaptation was present in only a minority of episodes within each type of variant (*Figure 7—figure supplement 2*). With a definition of adaptation based on a centrality value of more than 0, we found that the proportion of adaptive episodes was 43, 20, and 22% with type C>C, C>I, and I>I variants, respectively. In addition, certain clinical syndromes were more strongly associated with adaptation. Within the colonising population (type C>C variants), almost 80% of cystic fibrosis episodes were adaptive, as opposed to one third of episodes of skin colonisation in atopic dermatitis (*Figure 7AB*). This is consistent with within-host evolution studies showing strong convergent evolution signals among bacterial populations colonising individuals with cystic fibrosis, not only in case of *S. aureus* colonisation (*Long et al., 2020*) but also *P. aeruginosa* (*Marvig et al., 2015*) and *Mycobacterium abscessus* (*Bryant et al., 2021*); however, one study found adaptive evolution signals in atopic dermatitis (*Key et al., 2021*). We also observed that adaptation among infection episodes correlated with infection extent. Episodes of infective endocarditis episodes displayed higher adaptation metrics (46% with centrality >0) than bacteraemia with additional infection foci (28%) and bacteraemia without focus (17%) (*Figure 7D–E*).

To explore the syndrome-specificity of adaptation signatures, we mapped mutations in the most significantly enriched adaptive genes to clinical syndromes of colonisation and infection (*Figure 7* panels C and F). As expected, syndromes with high prevalence of adaptation had higher numbers of episodes with adaptive mutations; however, some genes appeared to be preferentially mutated. For example, *rpsJ*, *stp1*, and SAUSA300_1230 were over-represented in cystic fibrosis, while no clear pattern of mutations was discernible for nasal carriage episodes. Within infection syndromes, *mprF* and *purR* mutations were more prevalent in endocarditis, and *yjbH* mutations were only found in severe infections (bacteraemias with additional foci and endocarditis). Some genes appeared to be distinctive for low adaptation groups (atopic dermatitis and skin infections); however, the low number of adaptative mutations prevented an accurate assessment of these profiles.

## Discussion

Within-host evolution of bacterial pathogens such as *S. aureus* is thought to be governed by a combination of positive selection for variants that confers an advantage within the host and random fixation of mutations (genetic drift) (*Klemm et al., 2016*; *Didelot et al., 2016*). Sudden changes in the effective population size (bottlenecks) can cause genetic drift, for example, when a single or few bacterial cells invade the bloodstream or when a secondary infection foci is established in tissues and organs. Consistent with this view, animal studies have shown that after infecting the blood with a polyclonal population, bacteraemic infection is established stochastically by a single clone (*McVicker et al., 2014*; *Guerillot et al., 2018*), although estimating the bottleneck size at invasion from clinical sequences has been more challenging (*Abel et al., 2015*). On the other hand, several lines of evidence support the role of positive selection and adaptive evolution during *S. aureus* infection. First, adaptive phenotypic features appear to be acquired during infection. The most obvious adaptative

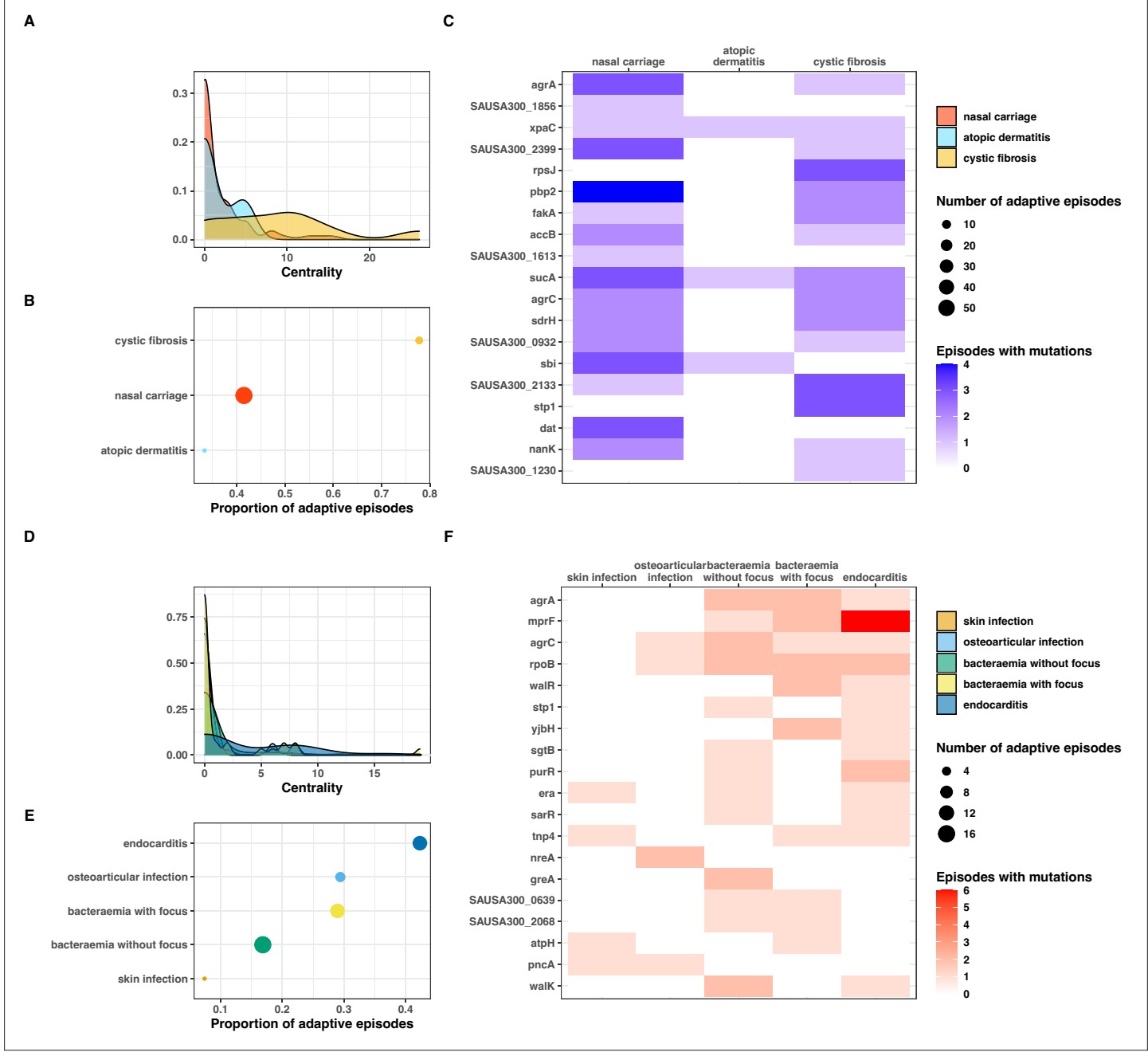

**Figure 7.** Clinical correlates of adaptive signatures within colonising (colonising-colonising [type C>C,] panels A–C) and invasive (invasive-invasive [type I>I], panels D–F) bacterial populations. Adaptation was inferred by computing the Jaccard index of shared mutated genes between independent episodes, followed by network analysis of infection episodes pairs. The node centrality measure was used as an indicator of adaptation. To avoid overinflation of mutated genes, the calculation was limited to the 20 most significantly enriched genes within each group of mutations. (**A, D**) Density of centrality values across colonisation (panel A) and infection categories (panel D). (**B, E**) Number and proportion of adaptive episodes. An adaptive episode was defined by a centrality >0. (**C, F**) Distribution of mutations in the 20 most significantly enriched genes across categories of colonisation (panel C) and infection (panel F).

The online version of this article includes the following figure supplement(s) for figure 7:

**Figure supplement 1.** Clinical manifestations and infection sites of invasive episodes, grouped by the infection syndromes classification used for the adaptation analysis.

**Figure supplement 2.** Network of colonisation/infection episodes for colonising-colonising (type CC) (panel A), colonising-invasive (type CI) (panel B), and invasive-invasive (type II) variants (panel C).

phenotype is secondary resistance to antistaphylococcal antibiotics such as rifampicin, vancomycin (*Howden et al., 2006*), daptomycin (*Peleg et al., 2012*), and oxacillin (*Giulieri et al., 2020*). Crucially, these resistance phenotypes can be associated with pleiotropic, pathoadaptative phenotypes such as small colony variant and immune evasion (*Guérillot et al., 2018*; *Jiang et al., 2019*; *Guérillot et al., 2019*). Furthermore, phenotypic adaptation (e.g. loss of toxicity) has been observed upon transition from colonisation to infection (*Laabei et al., 2015*), supporting the concept that invasive infection is linked to pathoadapted strains. At the molecular level, an excess of protein-truncating mutations in invasive strains (*Giulieri et al., 2018*) and in late colonising strains leading to infection (*Young et al., 2012*) have been noted. While this observation alone could be explained by relaxed constraint resulting from reduced population size (*Didelot et al., 2016*), it has been suggested that loss of gene function might be a common adaptation mechanism of within-host evolution (*Gatt and Margalit, 2021*), as supported by evidence of gene- or pathway-specific enrichment of mutations across independent infection episodes (*Young et al., 2017*).

Despite support for adaptive evolution from previous studies, it has been difficult to identify specific molecular signatures of adaptation during infection, due to the limited power of previous within-host studies of bacteraemia and other serious *S. aureus* infections that were often limited to a restricted number of episodes. To increase our ability to identify signatures of adaptation and find significantly enriched loci, we analysed multiple sources of genetic variation (point mutations, large deletions, IS insertions, and copy number variants) in a large collection of independent episodes of *S. aureus* colonisation and infection from 25 studies. We predicted that the main advantage of our approach would be to increase the ability of detecting convergence of genetic variants arising during invasive infections as opposed to those detected during the colonisation and upon transition from colonisation to infection. To test this hypothesis, we classified within-host variants based on their likely position in the within-host phylogeny (*Figure 1D*).

Bacterial adaptation is promoted by genomic plasticity; however, within-host evolution is characterised by low genetic variation (*Giulieri et al., 2018*). Based on our previous genomic studies of *S. aureus* bacteraemia, we reasoned that chromosome structural variants may provide an additional mechanism to increase genetic variation during infection. Here, we found that new insertions of IS are strongly enriched during invasive infection. However, despite this 20-fold enrichment, IS insertions remained a rare source of variation even within invasive strains and appeared to have a selective contribution to adaptation (i.e. limited to specific loci such as *agrC*). Similarly, large deletions and copy number variants appeared to play a less prominent role in adaptation, although we did not include them in our enrichment analysis.

Together with the enrichment for LOF mutations, which is another feature of evolution within the invasive population in our analysis, these IS movements suggest that a pattern of reductive evolution (genome degradation through loss of genes or accumulation of LOF mutations) emerges during within-host evolution of invasive *S. aureus*. This genome degradation might be related to less effective purifying selection (loss of deleterious alleles) in the invasive population due to a decrease in effective population size and a shorter evolutionary timescale (*Didelot et al., 2016*). However, our data indicate that these changes converge to specific genes, operons, and pathways, suggesting an adaptative benefit. Reductive evolution has been described in several 'commensal-to-pathogen' settings (*Toft and Andersson, 2010*). Although niche adaptation through reductive evolution has been described in extracellular pathogens (*Stinear et al., 2007*), a smaller (reduced) genome is a hallmark of obligate intracellular endosymbiotic bacteria (*Batut et al., 2014*). Since it appears that invasive *S. aureus* is able to reside intracellularly (promoting dissemination through mobile phagocytes during *S. aureus* sepsis [*Thwaites and Gant, 2011*; *Surewaard et al., 2016*], and immune evasion), it is plausible that this pattern of reductive evolution reflects an adaptation of invasive *S. aureus* to an invasive and intracellular lifestyle, as it has been shown for other facultative intracellular pathogens such as non-typhoidal *Salmonella* (*Klemm et al., 2016*). However, it is possible that these signatures of reductive evolution might be only temporary, as genome degradation might be present only in a minority of strains or be reversible; moreover, LOF mutations are expected to be more likely than gain of function mutations.

Beside reductive evolution, another distinctive feature of within-host evolution during invasive infection was intergenic mutations (both point mutations and IS insertions). In a within-host evolution study of *S. pneumoniae* colonisation, it was shown that intergenic sites were under convergent evolution (*Chaguza et al., 2020*). Mutations in promoter sequences of some core genes can play an

important role in antibiotic resistance as it was repeatedly shown for *pbp4* and resistance to beta-lactam antibiotics (*Basuino et al., 2018*). The role of intergenic mutations in within-host evolution was shown in a study of *P. aeruginosa* infection, where convergent evolution targeted several intergenic regulatory regions including upstream of antibiotic resistance genes (*Khademi et al., 2019*).

Previous work on within-host evolution by our group (included in this analysis) has established that *agrA* mutations are significantly enriched upon transition from colonisation to infection (*Young et al., 2017*). In addition, we have shown through genomics and targeted mutagenesis that mutations in key genes such as *walKR* (*Howden et al., 2011*) and *rpoB* (*Gao et al., 2013*) play a key pathoadaptive role in selected cases of persistent *S. aureus* infections. In this study, we increased our ability to discover potential targets of adaptation by analysing several mechanisms of genetic variation and applying several layers of annotation. As compared to previous work on *S. aureus*, this approach provides a higher-resolution picture of within-host evolution and adaptation. Importantly, this analysis remains robust after removing more than 1000 sequences from the largest within-host study included (*Figure 3—figure supplement 7*). We increase here the generalisability of our findings. We expand the list of genes targeted by convergent evolution and show that there are distinctive adaptation pathways in colonising and invasive populations. We confirm that the dominantly mutated loci belonged to the *agr* locus, in particular *agrA* and *agrC*. This finding is consistent with a large body of literature that predated the genomic era (*Novick and Geisinger, 2008*) that supports the role of the *agr* locus as the master regulator of gene expression in *S. aureus*. *Agr*-mediated adaptation was so important that we found a highly significant enrichment of *agr* mutations arros all type variants, including within colonising strains (type C>C variants). Shopsin et al. showed that ~10% of healthy *S. aureus* carriers held an agr-defective strain and that prior hospitalisation was significantly associated with agr-defective status, suggesting prior adaptive pressures (*Shopsin et al., 2008*).

Consistent with the distinctive general patterns of evolution displayed during invasive infection, some genes and loci were specifically mutated within invasive strains. Some of these genes were emerging targets of *S. aureus* pathogenesis in vivo, such as *purR* and *yjbH* that were not singled out in previous within-host evolution investigations. Others were known determinants of antibiotic resistance, including *mprF*, *rpoB,* and the *walKR* operon. This underscores the crucial role of antibiotic exposure in shaping adaptive evolution during invasive infection. A recent study of within-host evolution during cystic fibrosis found that resistance genes were hotspots of convergent evolution in this population, which is frequently treated with antibiotics and shows features of phenotypic adaptation (*Long et al., 2020*).

While most mutations in adaptive loci were substitutions within the coding sequence, about 40% of *walKR* operon mutations were located outside of the coding regions of *walR* and *walk*, emphasising the need to study intergenic mutations and mutations throughout an operon to capture adaptive signatures. This observation highlights the importance of expanding the analysis of intergenic mutations for the detection of adaptive mutations, in particular those linked to antibiotic resistance.

If within-host evolution represents adaptative evolution, it is possible that adaptation involved an accumulation of mutations and possibly epistatic mechanisms. Our data show that some mutations are specific for invasive strains; these mutations may reflect late adaptation, occurring after evolution during colonisation in the nose and upon transition from colonisation to infection, and thus occurring after adaptive mutations were acquired during earlier stages. This evolutionary pattern of stepwise adaptation (or adaptive continuum) encompassing the entire within-host evolutionary arch has been well described for cancer (*Abbosh et al., 2017*) and has been also investigated in a study of transition from colonisation to infection (*Young et al., 2012*). One way to capture this is mutation co-occurrence analysis. Here, we show mutation co-occurrence within the *agr* operon but also co-occurrence of the same mutations in two uncharacterised proteins in *S. aureus*. Our network of mutation co-occurrence linked to *agrAC* mutations might suggest a potential pathway of stepwise adaptation following initial mutations acquisition in the *agr* locus, an hypothesis that has been explored in one of studies included in our analysis (*Altman et al., 2018*).

While combining multiple studies allowed us to increase statistical power in order to detect genome-wide convergent signals of adaptation, this approach has some limitations. The quality of the publicly available sequences and metadata can be heterogeneous, despite performing quality control assessment, for example, due to different read coverage across studies. Lack of consistent metadata might have impacted the clinical categorisations used here, including the distinction between

colonising and invasive strains (however, 83% of the strains could be classified unambiguously based on the site of collection). In addition, detection of structural variants from short reads is not as accurate as from long reads; for example, chromosomal inversions can be missed if their inversion site span is larger than the insert site of the paired-end reads (*Guérillot et al., 2019*). Furthermore, the majority of the episodes included had low sampling density. More contemporary strategies leverage a multiplicity of samples and deep sequencing strategies to capture within-host diversity, allowing to detect minority variants that could be relevant for adaptation and to obtain a more accurate understanding of evolutionary dynamics within the host, including estimating bottlenecks (*Hall et al., 2019*) and tracking the fall and rise of within-host lineages (*Bryant et al., 2021*). Finally, while several adaptive loci identified here have been previously assessed experimentally, the functional impact of adaptive variants in less characterised genes and intergenic mutations warrants further exploration using targeted mutagenesis (*Monk et al., 2015*).

Ultimately, the goal of detecting adaptive signals is to identify new mechanisms of pathogenesis or resistance to therapeutic targets and to inform prediction of clinical outcomes. So far, studies have failed to show consistent associations between specific clinical outcomes and genetic features of infecting (or colonising) *S. aureus* strains. This might be related to the dominance of host/environmental factors, but it could also be linked to the limited power of studies performed so far. By contrast, within-host evolution studies are an elegant approach to identify signatures of adaptation that might be candidate markers of important clinical outcomes, such as infection risk in case of colonisation or treatment failure in case of infection. Here, we show that adaptation signatures are at least partially specific to colonisation, infection, and upon transition from colonisation to infection and that adaptive changes are more frequent in distinctive infection episodes (complicated bacteraemia and endocarditis). These findings suggest that adaptive signatures might be indicative of important clinical outcomes. In the future, precision medicine in infectious diseases could follow the lead of cancer genomics, where within-host evolution studies have tracked the evolution of cancer clones and enabled the detection of high risk mutations early.

## Materials and methods

### Literature search

We conducted a search of articles indexed in PubMed before the 11 August 2020 using the keyword 'aureus' in combination with either 'genomics' or 'whole genome sequencing' and with either 'within-host evolution', '*in vivo* evolution', 'adaptation', or 'bacteraemia'. The records retrieved through this search were combined with additional citations identified through other sources. After removing duplicates, this resulted in 815 citations that were screened based on following inclusion criteria: (i) whole-genome sequencing of human *S. aureus* isolates; (ii) >1 *S. aureus* isolates sequenced per individual; (iii) sequences (reads or assemblies) publicly available; and (iv) minimum sequences metadata available (either with the manuscript or linked to the sequences): patient ID, date of collection (or collection interval in reference to a baseline isolate), and source of the sample. After excluding studies not satisfying the inclusion criteria (730 based on the title, 46 based on the abstract, and 15 after reviewing the full text), we kept 24 within-host evolution studies.

### Extraction of sequence metadata

For each of the included studies, the following variables were extracted either from reads, metadata (when available) or from the publication/supplementary data: identifier linking the sequences to a patient or an episode of infection, date of collection (when available) or collection interval in reference to a baseline isolate, and site of collection of the isolate. Isolates were broadly categorised in 'colonising' and 'invasive' based on the site of collection, when the information was unambiguous (e.g. 'nose' for 'colonising' or 'blood' for 'invasive'). When the information on the body site was not sufficient (e.g. 'skin' or 'lung'), the categorisation was based on further details provided in the publication. When available, phenotypic metadata and antibiotic treatments were also extracted from the publication. We used clinical and site data to classify colonisation episodes in 'nasal carriage', 'atopic dermatitis', and 'cystic fibrosis' and infection episodes in 'skin infection' (skin infection site surgical site infection without other foci), 'osteoarticular infection' (bone/join infection without other foci), 'bacteraemia without focus' (bloodstream infection, no other foci, and expect for vascular catheter or skin),

'bacteraemia with focus' (bloodstream infection with other focus involving the lung, nervous system, bone and joints, or internal organs), and 'endocarditis' (based on diagnosis reported in the publication or in the clinical metadata).

## Sequence processing

Sequences (reads and assemblies) and metadata were downloaded from the European Nucleotide Archive and the National Center for Biotechnology Information (NCBI), respectively using the BioProject accession or the genome accession. Quality assessment of the reads was performed by calculating mean read depth and the fraction of *S. aureus* reads using Kraken 2, v2.0.9-beta (*Wood et al., 2019*) and by extracting metrics from reads assemblies constructed using Shovill, v1.1.0 (https://github.com/tseemann/shovill, *Seemann, 2022c*) and annotated using Prokka, v1.14.6 *Seemann, 2014* . ST was inferred from the assembly using Mlst, v2.19.0 (https://github.com/tseemann/mlst, *Seemann, 2022b*), and resistance genes were detected using Abricate, v1.0.1 (https://github.com/tseemann/abricate, *Seemann, 2022a*). Reads were discarded if the mean coverage depth was below 35, the majority of reads were not *S. aureus,* or the size of the assembly was below 2.6 megabases. Assemblies downloaded from the NCBI repository were discarded if the genome size was below 2.6 megabases.

## Sequences from the CAMERA2 trial

We collected *S. aureus* strains from bacteraemia episodes included in the CAMERA2 trial (Combination Antibiotics for Methicillin Resistant *S. aureus*), where at least two strains per episode were available. The CAMERA2 trial was performed between 2015 and 2018 in Australia, New Zealand, Singapore, and Israel and randomised participants with methicillin-resistant *S. aureus* bacteraemia to either monotherapy with vancomycin or daptomycin or combination therapy with vancomycin or daptomycin plus an antistaphylococcal beta-lactam (flucloxacillin, cloxacillin, or cefazolin) (*Tong et al., 2020*). Strains were isolated from –80C glycerol onto horse-blood agar. Species were confirmed using matrix-assisted laser desorption/ionization time-of-flight mass spectrometry. Bacterial whole-genome sequencing was performed from single colonies on the Illunina NextSeq platform. Reads were checked for quality, assembled, and annotated as described above.

## Global phylogeny

To generate a global alignment of all sequences, reads and shredded assemblies were mapped to reference genome USA300 FPR3757 (assembly accession: GCF_000013465.1) (using Snippy, v4.6.0) (https://github.com/tseemann/snippy; *Seemann, 2022d*). The core genome alignment was obtained using Snippy; sites with >10% gaps were removed using Goalign (*Lemoine and Gascuel, 2021*) and constant sites were removed using SNP-sites (*Page et al., 2016*), for a final length of 186,825 bp. A maximum-likelihood phylogenetic tree of 2590 sequences (those kept in the analysis after excluding genetically unrelated strains, see below) was inferred using IQ-TREE, v2.0.3 *Minh et al., 2020*.

## Variant calling

We have previously shown that the accuracy of variant calling in within-host evolution analyses is improved when mapping reads to an internal draft assembly as opposed to a closely related closed genome (*Giulieri et al., 2018*). Here, we applied the same approach, where we selected the internal reference among the sequences from the same patient or episode. When available, the oldest colonising strain was selected. When only sequences from invasive strains were available, the oldest strain (baseline strain) was selected. When multiple sequences were available per sample (e.g. multiple colonies sequenced per plate) or at the same date, the reference was randomly selected among them. Snippy with default parameters (minimum reads coverage 10, minimum read mapping quality 60, and minimum base quality 13) was used for variant calling. To further improve the accuracy of the calls, we masked variants called from reference reads and those at positions where reference reads had a coverage below 10 (using the BEDTools suite [*Quinlan and Hall, 2010*]).

## Filtering of genetically unrelated sequences

The threshold for removing genetically unrelated sequences was set empirically at 100 episode-specific variants based on the upper Tukey's fence of the distribution of the number of variants in same-episode isolates belonging to the same ST (*Figure 2—figure supplement 1*).

## Estimation of within-host mutation rates

To estimate within-host mutation rates within colonising and invasive populations, a linear regression was fitted to model the relationship between sampling time (in years after the first isolate) and number of mutations relative to the internal reference. Only episodes with at least two strains collected at least one day apart were included in this analysis. The mutation rate μ was computed as follows μ = β/g, where β is the regression parameter and g is the mean genome size of the internal references (2.79 Mb). Regression diagnostics were performed using the R package performance (*Lüdecke et al., 2021*).

## Detection of chromosome structural variants

Using BWA-MEM (*Li, 2013*), reads and shredded contigs were aligned to the closest available complete genome (either internal to the dataset or selected from the NCBI repository based on the mash distance). To detect large deletions ( ≥ 500 bp), reads coverage was computed using BEDTools, as described in *Giulieri et al., 2018*. To detect new IS insertions, split reads were extracted, filtered, and annotated as described in *Giulieri et al., 2018*. We used the R package CNOGpro (*Brynildsrud et al., 2015*) to detect 1000 bp windows with an estimated copy number above one as compared to the internal reference. The package calculates the reads coverage per sliding windows of the chromosome, performs a G+C bias normalisation, and infers copy number state using a Hidden Markow Model. We ran the package with default parameters, with the exception of the length of the sliding window that was set at 1000 bp. For each class of structural variant and within each episode, we used BEDTools to mask regions where the variant was already present in internal reference.

## Prediction of functional impact of variants

Functional impact of variants was extracted from the Snippy output, which uses SnpEff to infer the functional effect of the detected mutations (*Cingolani et al., 2012*). SnpEff categories for coding regions were aggregated in 'truncating' (frameshift, stop codons, and start codons), 'non-synonymous substitutions', and 'synonymous substitutions'. Non-synonymous substitutions were further investigated using PROVEAN, v1.1.5 (58) using the non-redundant protein database provides on the PROVEAN repository (ftp://ftp.jcvi.org/pub/data/provean/nr_Aug_2011/). Substitutions were classified as 'deleterious' if the PROVEAN score was –2.5 or less and 'neutral' otherwise.

## Internal variant annotation

To ensure a consistent annotation of mutated genes across independent episodes, we clustered amino-acid sequences using CD-HIT, v4.8.1 with an identity threshold of 0.9. The BEDTools suite was used to annotate mutated intergenic regions with upstream and downstream coding regions and the distance separating the mutation from the start or the end of the gene. For the operon analysis, intergenic mutations were classified according to their location within a presumed promoter based on blasting the sequence of unique promoters (as determined in [*Prados et al., 2016*]) on the draft assembly of the internal reference. Phage genes were annotated using blastp and the PhageWeb database (http://computationalbiology.ufpa.br/phageweb/).

## Variant annotation using reference strain FPR3757

To compare mutated genes across separated episodes, we used blastp to identify homologues of each CD-HIT cluster of mutated genes in USA300 FPR3757. Genes in FPR3757 were further annotated using the database provided in the AureoWiki repository (*Fuchs et al., 2018*), and operon annotations of FPR3757 were retrieved from Microbes Online (*Dehal et al., 2010*). In addition, we used the text mining tool PaperBLAST to search for publications containing data on homologues of uncharacterised FPR3757 proteins (*Price et al., 2017*). Only protein-altering variants in genes with FPR3757 homologues (excluding plasmid genes and phage genes) were kept for the analysis of convergence at gene and operon level and the gene enrichment analysis.

## Classification of variants

Mutational and structural variants were classified in to type C>C (within colonising strains), type C>I (between colonising and invasive strains), and type I>I (within invasive strains) as follows: all variants arising in colonising strains were classified as type C>C, while variants among invasive strains were classified as type C>I if they were found in a baseline invasive strain (defined as the oldest invasive strain; when multiple sequences were available at same time, the baseline invasive strain was selected randomly), and as type I>I if they were found between invasive strains but not on the baseline invasive strain. This approach is based on the assumption that co-infection or superinfection is rare, as we have shown previously for bacteraemia (*Giulieri et al., 2018*).

## Calculation of the Neutrality Index (NI)

A modified McDonald-Kreitman table was compiled a described in *Stoletzki and Eyre-Walker, 2011*, where a ratio was calculated between non-synonymous, protein-truncating, IS insertions, intergenic and deletion variants, and synonymous variants. The NI was obtained by dividing the ratio calculated above for type C>I and type I>I by the ratio for type C>C variants that were used as reference group. Significance was tested by Fisher's Exact test.

## dN/dS analysis

We used the R package dNdScv (*Martincorena et al., 2017*) to obtain dN/dS ratios for non-synonymous mutations, indels, and missense mutations (stop codons) for all FPR3757 genes, based on variants called when mapping all reads on FPR3757 and after subtracting variants from the internal reference reads and variants in positive where internal reference reads had a low coverage. Since this analysis could be hampered by potential false-positive variants resulting from mapping reads on a single reference (*Giulieri et al., 2018*), we also used our curated list of within-host mutations obtained from episode-specific variant calling to calculate crude dN/dS ratios by dividing the number of protein modifying mutations by the number non-synonymous mutations and computed p values by Fisher exact test as in *Long et al., 2020*.

## Gene and operon enrichment analysis

We calculated the enrichment of protein-altering mutations across all coding regions of FPR3757 (excluding plasmid genes and phage genes) using the approach described in *Young et al., 2017*. The variant enrichment per gene $i$ was calculated as follows: $(N_i/L_i)/(\Sigma_n/\Sigma_l)$, where $N_i$ is the number of variants per gene $i$, $L_i$ is the length of gene $i$, $\Sigma_n$ is the total number of variants, and $\Sigma_l$ is the total length of the genes. We used Poisson regression to model the number of variants per gene $j$ under the null hypothesis (no enrichment), as defined by the equation $\lambda_0 L_j$, where $\lambda_0$ is the expected number of variants in any gene and $L_j$ is the gene length. Under the alternative hypothesis (enrichment of variant in gene $i$), the estimated number of variants is $\lambda_i L_i$ for gene $i$, and $\lambda_1 L_j$ for any other gene $j$. The model parameters $\lambda_0$, $\lambda_1$, and $\lambda_l$ were obtained using maximum likelihood and tested for significance using the likelihood ratio test. The genome-wide significance cut-off was calculated using the Bonferroni correction (0.05 divided by the number of unique genes or operons) and the suggestive significance cut-off (1 divided by the number of unique genes or operons), as implemented for bacterial genome-wide associated studies in *Lees et al., 2017a*.

## Gene set enrichment analysis

We used the PANNZER platform (*Törönen et al., 2018*), to retrieve a gene ontology annotation of FPR3757 based on the GO terms. We modified the 'antibiotic response' category by adding a curated list of antibiotic resistance genes downloaded from the NCBI Anti-Microbial Resistance (AMR) gene reference database (*Feldgarden et al., 2019*). The GSEA was performed as implemented in the R package clusterProfile (*Yu et al., 2012*). Genes with a FPR3757 homologue were ranked according to the significance of the enrichment of protein-modifying mutations (gene enrichment analysis, see above), and the GSEA was carried out with a minimum gene set size of 10 and using the FDR method for adjustment for multiple testing.

## Mutation co-occurrence analysis

To detect co-occurrence of mutations and mutated genes across independent episodes, we constructed a co-occurrence matrix using the R package co-occur (*Griffith et al., 2016*). A

co-occurrence of mutations or mutated genes in at least two independent episodes was interpreted as convergent and as a sign of potential epistatic interaction. The network of co-occurrence of mutated genes was visualised using the R package ggraph (https://cran.r-project.org/web/packages/ggraph/index.html).

## Network analysis of adaptation signatures

The pairwise calculation of the Jaccard index between set of mutated genes was performed in R. The calculations were performed both with the entire set of mutated FPR3757 genes and with the 20 most significantly enriched genes in each group of variants. A network of shared mutated genes between independent episodes was constructed using ggraph, where edges represented episode connections based on the Jaccard index. We used the R package tidygraph to extract the node centrality (function centrality_degree) as a summary measure of the degree of adaptation of the episodes. The network graph and analysis were performed for each group of variants separately.

## Acknowledgements

The authors declare that they have no competing interests. All data needed to evaluate the conclusions in the paper are present in the paper and/or the Supplementary Materials. This work was supported by a Research Fellowship to BPH and TPS from the National Health and Medical Research Council, Australia. SGG was supported by a PhD scholarship of the University of Melbourne. We acknowledge the CAMERA2 Study Group for sharing sequences and clinical metadata of trial participants with multiple sequential bacteraemia strains: Nick Anagostou, David Andresen, Sophia Archuleta, Narin Bak, Alan Cass, Mark Chatfield, Alan Cheng, Jane Davies, Joshua Davis, Yael Dishon, Ravindra Dotel, Patricia Ferguson, Hong Foo, Vance Fowler, Niladri Ghosh, Timothy Gray, Stephen Guy, Natasha Holmes, Benjamin Howden, Sandra Johnson, Shirin Kalimuddin, David Lye, Stephen McBride, Genevieve McKew, Niamh Meagher, Jane Nelson, Matthew O'Sullivan, David Paterson, Mical Paul, David Price, Anna Ralph, Matthew Roberts, Owen Robinson, Ben Rogers, Naomi Runnegar, Simon Smith, Archana Sud, Steven Tong, Adrian Tramontana, Sebastian Van Hal, Genevieve Walls, Morgyn Warner, Dafna Yahav, and Barnaby Young.

## Additional information

### Funding

| Funder | Grant reference number | Author |
|---|---|---|
| National Health and Medical Research Council | GNT1105525 | Timothy P Stinear |
| National Health and Medical Research Council | GNT1196103 | Benjamin P Howden |

The funders had no role in study design, data collection and interpretation, or the decision to submit the work for publication.

### Author contributions

Stefano G Giulieri, Conceptualization, Data curation, Formal analysis, Methodology, Visualization, Writing – original draft, Writing – review and editing; Romain Guérillot, Conceptualization, Formal analysis, Methodology, Software, Supervision, Writing – review and editing; Sebastian Duchene, Resources, Software, Writing – review and editing; Abderrahman Hachani, Investigation, Writing – review and editing; Diane Daniel, Data curation, Investigation, Project administration; Torsten Seemann, Methodology, Software, Supervision; Joshua S Davis, Investigation, Resources, Writing – review and editing; Steven YC Tong, Conceptualization, Investigation, Resources; Bernadette C Young, Methodology, Writing – review and editing; Daniel J Wilson, Methodology, Software, Writing – review and editing; Timothy P Stinear, Conceptualization, Methodology, Supervision, Writing – review and editing; Benjamin P Howden, Conceptualization, Funding acquisition, Supervision, Writing – review and editing

**Author ORCIDs**
Abderrahman Hachani (ID) http://orcid.org/0000-0001-8032-2154
Steven YC Tong (ID) http://orcid.org/0000-0002-1368-8356
Bernadette C Young (ID) http://orcid.org/0000-0001-6071-6770
Daniel J Wilson (ID) http://orcid.org/0000-0002-0940-3311
Timothy P Stinear (ID) http://orcid.org/0000-0003-0150-123X
Benjamin P Howden (ID) http://orcid.org/0000-0003-0237-1473

**Ethics**

Ethics approval was obtained at each partecipating site to the CAMERA2 trial and written informed onsent was obtained from each participant or surrogate decision maker.

**Decision letter and Author response**

Decision letter https://doi.org/10.7554/eLife.77195.sa1
Author response https://doi.org/10.7554/eLife.77195.sa2

---

# Additional files

**Supplementary files**

• Supplementary file 1. List of colonisation/infection episodes included with publication data (first author, year, PubMed id), number of strains, sites of collection, clinical characterstics, classification of colonisation, and infection episodes.

• Supplementary file 2. List of strains included with site and date of collection, sequence type, presence of the *mecA* gene, information on whether the strain was designed as internal reference or baseline index strain, mash distance to the internal reference, number of variants called (as compared to the internal reference), and sequencing metrics.

• Supplementary file 3. List of variants identified annotated with gene, gene sequence, FPR3757 homologue, and FPR3757 operon. Point mutations, insertion sequences insertions, large deletions, and copy number variants are presented separately.

• Supplementary file 4. Gene enrichment analysis for all mutated genes with a FPR3757 homologue with number of mutations, gene length, mutation enrichment, and p value based on a Poisson regression to model the number of variants per gene. Results are presented separately for the complete dataset and for colonising-colonising (type C>C), colonising-invasive (type C>I), and invasive-invasive (type I>I) variants.

• Supplementary file 5. Operon enrichment analysis for all FPR3757 operons (i.e. mutated genes that could be assigned to a FPR3757 operon) with number of mutations, operon length, mutation enrichment, and p value based on a Poisson regression to model the number of variants per operon. Results are presented separately for the complete dataset and for colonising-colonising (type C>C), colonising-invasive (type C>I), and invasive-invasive (type I>I) variants.

• Supplementary file 6. Gene set enrichment analysis for mutations in genes aggregated in gene ontologies (GO) categories with enrichment score, normalsied enrichment score (NES), and unadjusted and false-discovery rate (FDR) adjusted p value. Results are presented separately for the complete dataset and for colonising-colonising (type C>C), colonising-invasive (type C>I), and invasive-invasive (type I>I) variants.

• MDAR checklist

**Data availability**

All data generated or analysed during this study are included in the manuscript and supporting file 1-6. The code to call, filter and annotated within-host variants and to perform the enrichment analysis is available on github at https://github.com/stefanogg/staph_adaptation_paper, (copy archived at swh:1:rev:6ec5132855405e1c759fedadb4c70e295c1c1974).

The following dataset was generated:

| Author(s) | Year | Dataset title | Dataset URL | Database and Identifier |
|---|---|---|---|---|
| Giulieri SG, Daniel D, Seemann T, Davis JS, Tong SYC, Stinear TP, Howden BP | 2022 | Within-host evolution analysis of *S. aureus* bacteraemias included in the CAMERA-2 trial | https://github.com/ stefanogg/staph_ adaptation_paper | European Nucleotide Archive, PRJEB50796 |

The following previously published datasets were used:

| Author(s) | Year | Dataset title | Dataset URL | Database and Identifier |
|---|---|---|---|---|
| Gao W, Monk IR, Tobias NJ, Gladman SL, Seemann T, Stinear TP, Howden BP | 2015 | Large tandem chromosome duplications facilitate niche adaptation during persistent infection with drug-resistant *Staphylococcus aureus* | https://www.ncbi.nlm. nih.gov/bioproject/ PRJEB9193 | NCBI BioProject, PRJEB9193 |
| Howden BP | 2011 | Evolution of Multidrug Resistance during *Staphylococcus aureus* Infection Involves Mutation of the Essential Two Component Regulator WalKR | https://www.ncbi.nlm. nih.gov/sra/?term= SRA027352 | NCBI Sequence Read Archive, SRA027352 |
| Howden BP | 2011 | Evolution of Multidrug Resistance during *Staphylococcus aureus* Infection Involves Mutation of the Essential Two Component Regulator WalKR | https://www.ncbi.nlm. nih.gov/bioproject/ PRJNA29567 | NCBI BioProject, PRJNA29567 |
| Howden BP, McEvoy CR, Allen DL, Chua K, Gao W, Harrison PF, Bell J, Coombs G, Bennett-Wood V, Porter JL, Robins-Browne R, Davies JK, Seemann T, Stinear TP | 2011 | Evolution of Multidrug Resistance during *Staphylococcus aureus* Infection Involves Mutation of the Essential Two Component Regulator WalKR | https://www.ncbi.nlm. nih.gov/bioproject/ PRJNA29569 | NCBI BioProject, PRJNA29569 |
| Young BC, Golubchik T, Batty EM, Fung R, Larner-Svensson H, Votintseva AA, Miller RR, Godwin H, Knox K, Everitt RG, Iqbal Z, Rimmer AJ, Cule M, Ip CLC, Didelot X, Harding RM, Donnelly P, Peto TE, Crook DW, Bowden R, Wilson DJ | 2012 | Evolutionary dynamics of *Staphylococcus aureus* during progression from carriage to disease | https://www.ncbi.nlm. nih.gov/bioproject/ PRJEB2862 | NCBI BioProject, PRJEB2862 |
| Golubchik T, Batty EM, Miller RR, Farr H, Young BC, Larner-Svensson H, Fung R, Godwin H, Knox K, Votintseva A, Everitt RG, Street T, Cule M, CL Ip, Didelot X, Peto TE, Harding RM, Wilson DJ, Crook DW, Bowden R | 2013 | Within-host evolution of *Staphylococcus aureus* during asymptomatic carriage | https://www.ncbi.nlm. nih.gov/bioproject/ PRJEB2881 | NCBI BioProject, PRJEB2881 |

*Continued on next page*

*Continued*

| Author(s) | Year | Dataset title | Dataset URL | Database and Identifier |
|---|---|---|---|---|
| Burd EM, Alam MT, Passalacqua KD, Kalokhe AS, Eaton ME, Satola SW, Kraft CS, Read TD | 2014 | Development of oxacillin resistance in a patient with recurrent *Staphylococcus aureus* bacteremia | https://www.ncbi.nlm.nih.gov/bioproject/PRJNA248678 | NCBI BioProject, PRJNA248678 |
| Rishishwar L, Kraft CS, Jordan IK | 2016 | Population Genomics of Reduced Vancomycin Susceptibility in *Staphylococcus aureus* | https://www.ncbi.nlm.nih.gov/bioproject/PRJNA259799 | NCBI BioProject, PRJNA259799 |
| Trouillet-Assant S, Lelièvre L, Martins-Simões P, Gonzaga L, Tasse J, Valour F, Rasigade JP, Vandenesch F, Muniz Guedes RL, Ribeiro de Vasconcelos AT, Caillon J, Lustig S, Ferry T, Jacqueline C, Loss de Morais G, Laurent F | 2016 | Adaptive processes of *Staphylococcus aureus* isolates during the progression from acute to chronic bone and joint infections in patients | https://www.ncbi.nlm.nih.gov/bioproject/PRJNA298748 | NCBI BioProject, PRJNA298748 |
| Rouard C, Garnier F, Leraut J, Lepainteur M, Rahajamananav L, Languepin J, Ploy MC, Bourgeois-Nicolaos N, Doucet-Populaire F | 2018 | Emergence and Within-Host Genetic Evolution of Methicillin-Resistant *Staphylococcus aureus* Resistant to Linezolid in a Cystic Fibrosis Patient | https://www.ncbi.nlm.nih.gov/bioproject/PRJNA434495 | NCBI BioProject, PRJNA434495 |
| Langhanki L, Berger P, Treffon J, Catania F, Kahl BC, Mellmann A | 2018 | In vivo competition and horizontal gene transfer among distinct *Staphylococcus aureus* lineages as major drivers for adaptational changes during long-term persistence in humans | https://www.ncbi.nlm.nih.gov/bioproject/PRJEB22600 | NCBI BioProject, PRJEB22600 |
| Altman DR | 2018 | Genome Plasticity of agr-Defective *Staphylococcus aureus* during Clinical Infection | https://www.ncbi.nlm.nih.gov/bioproject/PRJNA393749 | NCBI BioProject, PRJNA393749 |
| Giulieri SG | 2018 | Genomic exploration of sequential clinical isolates reveals a distinctive molecular signature of persistent *Staphylococcus aureus* bacteraemia | https://www.ncbi.nlm.nih.gov/bioproject/PRJEB27932 | NCBI BioProject, PRJEB27932 |
| Benoit JB, Frank DN, Bessesen MT | 2018 | Genomic evolution of *Staphylococcus aureus* isolates colonizing the nares and progressing to bacteremia | https://www.ncbi.nlm.nih.gov/bioproject/PRJNA414752 | NCBI BioProject, PRJNA414752 |
| Suligoy CM, Lattar SM, Noto Llana M, González CD, Alvarez LP, Robinson DA, Gómez MI, Buzzola FR, Sordelli DO | 2018 | Mutation of Agr Is Associated with the Adaptation of *Staphylococcus aureus* to the Host during Chronic Osteomyelitis | https://www.ncbi.nlm.nih.gov/bioproject/PRJNA414566 | NCBI BioProject, PRJNA414566 |

*Continued on next page*

*Continued*

| Author(s) | Year | Dataset title | Dataset URL | Database and Identifier |
|---|---|---|---|---|
| Harkins CP, Pettigrew KA, Oravcová K, Gardner J, Hearn RMR, Rice D, Mather AE, Parkhill J, Brown SJ, Proby CM, Holden MTG | 2018 | The Microevolution and Epidemiology of *Staphylococcus aureus* Colonization during Atopic Eczema Disease Flare | https://www.ncbi.nlm.nih.gov/bioproject/PRJEB20148 | NCBI BioProject, PRJEB20148 |
| Tan X | 2019 | Chronic *Staphylococcus aureus* Lung Infection Correlates With Proteogenomic and Metabolic Adaptations Leading to an Increased Intracellular Persistence | https://www.ncbi.nlm.nih.gov/bioproject/PRJNA446073 | NCBI BioProject, PRJNA446073 |
| Loss G, Simões PM, Valour F, Cortês MF, Gonzaga L, Bergot M, Trouillet-Assant S, Josse J, Diot A, Ricci E, Vasconcelos AT, Laurent F | 2019 | *Staphylococcus aureus* Small Colony Variants (SCVs): News From a Chronic Prosthetic Joint Infection | https://www.ncbi.nlm.nih.gov/bioproject/PRJNA497214 | NCBI BioProject, PRJNA497214 |
| Kuroda M, Sekizuka T, Matsui H, Ohsuga J, Ohshima T, Hanaki H | 2019 | IS256-Mediated Overexpression of the WalKR Two-Component System Regulon Contributes to Reduced Vancomycin Susceptibility in a *Staphylococcus aureus* Clinical Isolate | https://www.ncbi.nlm.nih.gov/bioproject/PRJDB8056 | NCBI BioProject, PRJDB8056 |
| Azarian T, Ridgway JP, Yin Z, David MZ | 2019 | Long-Term Intrahost Evolution of Methicillin Resistant *Staphylococcus aureus* Among Cystic Fibrosis Patients With Respiratory Carriage | https://www.ncbi.nlm.nih.gov/bioproject/PRJNA527261 | NCBI BioProject, PRJNA527261 |
| Wüthrich D, Cuénod A, Hinic V, Morgenstern M, Khanna N, Egli A, Kuehl R | 2019 | Genomic characterization of inpatient evolution of MRSA resistant to daptomycin, vancomycin and ceftaroline | https://www.ncbi.nlm.nih.gov/bioproject/PRJNA488707 | NCBI BioProject, PRJNA488707 |
| Ji S, Jiang S, Wei X, Sun L, Wang H, Zhao F, Chen Y, Yu Y | 2020 | In-Host Evolution of Daptomycin Resistance and Heteroresistance in Methicillin-Resistant *Staphylococcus aureus* Strains From Three Endocarditis Patients | https://www.ncbi.nlm.nih.gov/bioproject/PRJNA577181 | NCBI BioProject, PRJNA577181 |
| Miller CR, Dey S, Smolenski PD, Kulkarni PS, Monk JM, Szubin R, Sakoulas G, Berti AD | 2020 | Distinct Subpopulations of Intravalvular Methicillin-Resistant *Staphylococcus aureus* with Variable Susceptibility to Daptomycin in Tricuspid Valve Endocarditis | https://www.ncbi.nlm.nih.gov/bioproject/PRJNA544229 | NCBI BioProject, PRJNA544229 |
| Petrovic Fabijan A, Lin RCY, Ho J, Maddocks S, Ben Zakour NL, Iredell JR | 2020 | Safety of bacteriophage therapy in severe *Staphylococcus aureus* infection | https://www.ncbi.nlm.nih.gov/bioproject/PRJNA541589 | NCBI BioProject, PRJNA541589 |

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
