## [Editor Report]

This study offers a comprehensive examination of *Staphylococcus aureus* evolution during infection. It provides a useful analysis of select genetic signatures during bacterial adaptation. A combination of multiple layers of genome annotation and point mutation variant detection compellingly supports the correlation of infection outcomes with adaptation signatures in *S. aureus*.

---

## [Decision Letter]

**Decision letter after peer review:**

Thank you for submitting your article "Niche-specific genome degradation and convergent evolution shaping *Staphylococcus aureus* adaptation during severe infections" for consideration by *eLife*. Your article has been reviewed by 2 peer reviewers, and the evaluation has been overseen by a Reviewing Editor and Bavesh Kana as the Senior Editor. The following individuals involved in the review of your submission have agreed to reveal their identity: Daria Van Tyne (Reviewer #1); Meiqin Zheng (Reviewer #2).

Essential revisions:

1. One weakness of the study is a lack of a classification of the variants detected in convergent loci. For example, which genes do the authors think are acquiring gain-of-function versus loss-of-function mutations? One other weakness is a lack of functional studies exploring some of the more novel signals detected (such as hypothetical proteins with "no data on S. aureus"). Can the authors comment or provide data on this?

2. Line 130-131: Was there any association between sampling frequency and variant accumulation in patients between early vs. late adapted populations? Can the available data be used to estimate mutation rates or bottleneck sizes in moving from colonization->early infection->late infection?

Other comments that must be addressed

1. Line 78: Reference 13 is a study in E. faecalis, not E. faecium. Chilambi et al., 2020 PNAS examine colonization vs. infection in VREfm.

2. Lines 143-145: Please define "early stage," "late infection stage," "persistence," and "treatment failure" here. These terms are not defined in the methods or anywhere else in the paper.

3. Line 153 should reference Figure 2B.

4. Line 183 should reference Figure 2 —figure supplement 2.

5. Lines 207-208: How many genes had a homolog in the reference strain? What percentage of the genome was this?

6. Was the definition of "colonization" vs. "infection" consistent across all studies compiled here? If not this should be acknowledged and discussed.

7. Figure 1: Panels B and D are somewhat confusing. Consider revising panel D, or moving it to the supplement.

8. Figure 3 and others like it: the triangles corresponding to "2" vs. "3" independent acquisitions do not look noticeably different on the figures.

9. Figure 3 —figure supplement 1: what does "ext*?" mean?

10. Line 416: what is the reference base of the definition about adaptation based on a centrality value of more than 0?

11. Line 712-714 and 722-723: These two sentences could be integrated into one.

12. In general, the article should be written with greater readability for the generalist audience of *eLife*.

13. Please check the whole article again and modify some mistakes such as Line 240, line 259, line 289, line 784, Figure 3 —figure supplement 5, and so on.

---

## [Author Response]

Essential revisions:1. One weakness of the study is a lack of a classification of the variants detected in convergent loci. For example, which genes do the authors think are acquiring gain-of-function versus loss-of-function mutations?

We agree with the reviewer that our classification system doesn’t account for the predicted functional impact of amino acid substitutions. This is especially important for convergent loci where most or all aggregated mutations are substitutions such as *mprF* and *sucA.* To address this shortcoming, we used PROVEAN to predict the impact of amino acid substitution on the protein function (lines 915-923). We used a δ score of -2.5 to differentiate between deleterious and neutral substitutions. The results are presented in Figure 3 – supplement 3 (new) and in the manuscript (lines 295-322). The output of PROVEAN is also provided with Supplementary file 3.

Based on *in silico* predictions (protein-truncations, low PROVEAN scores) we hypothesise that most convergent loci carry loss-of-function mutations. However, gain-of-function mutations are hard to predict from the sequence. We reviewed the literature on genes with low prevalence of truncations and higher PROVEAN scores and only confirmed gain-of-function mutations for *mprF*.

One other weakness is a lack of functional studies exploring some of the more novel signals detected (such as hypothetical proteins with "no data on S. aureus"). Can the authors comment or provide data on this?

Following the reviewer’s suggestion, we have performed a review of hypothetical proteins: we searched for homologs and functional domains and used PaperBlast to identify relevant papers (lines 947-948). We have revised tables 3/4 and added some relevant references that can provide more background information on the novel adaptive loci. We also agree with the reviewer that functional studies of candidate adaptive loci of unknown function will be important. Detection of previously described pathoadaptive loci such as *agr*, *stp1*, *saeR*, *vraR* and *walR* (Figure 3), underscores the validity of our approach. However, determining roles for genes in transition from colonising to invasive (and other pathoadaptive phenotypes) is non-trivial and of course represent substantial, stand-alone studies. We are planning to conduct these investigations and we hope that our work will encourage others to also dive into such studies (see also discussion, lines 750-754).

2. Line 130-131: Was there any association between sampling frequency and variant accumulation in patients between early vs. late adapted populations?

We assessed the correlation between number of samples and mean mutation counts per episode and stage of infection (colonisation, early, late). No correlation was found, except for early infections, where there was a weak association between number of sequences and mean mutation counts (R^2^ = 0.042, p = 0.001). This analysis is presented in Figure 2 —figure supplement 2 and mentioned in the manuscript (lines 172-175).

Can the available data be used to estimate mutation rates or bottleneck sizes in moving from colonization->early infection->late infection?

We investigated mutation counts and rates within colonising and invasive populations by fitting a linear regression of mutation counts in response to the collection time. The regression suggests that mutation rates are higher in the invasive population, however this analysis should be taken with caution given that we are using multiple studies and sampling strategies and that the models displayed heteroskedasticity and non-linear distribution (results, lines 177-184 and methods 886-893). As hypothesised by the reviewer, this is a consequence of the heterogenous sampling approaches across the studies included in the analysis. Because of these limitations, the regression is presented in the supplementary data (Figure 2 —figure supplement 3 and 4).

We agree with the reviewers’ that there is an interest in inferring bottleneck size at invasion from clinical data. Data from animal experiments suggest that there is a narrow bottleneck at the transition from colonisation to infection and upon organ seeding during bacteraemia. These analyses have used molecular markers and experimental design to infer the size of the bottleneck in animal models of bacteraemia and are therefore difficult to replicate with clinical sequences. Alternatively, the bottleneck size can be calculated using population genomics approaches, provided sufficient diversity and sampling depth is present, as it is the case in deep sequencing studies of viral infections. A few studies have also used deep sequencing approaches to calculate the size of the transmission bottleneck in bacterial infections. In a study by Hall *et al.,* (*eLife* 2019;8:e46402), the size of the transmission bottleneck was assessed using a Bayesian phylogenetic method in six colonisation pairs that fulfilled strict criteria in terms of statistical support and sampling depth (95% posterior trees and > 5 tips [i.e. individual sequences] supporting the direction of transmission). Our dataset does not provide sufficient sampling coverage to apply one of the above approaches. The rate of coinfections with genetically distant lineages can be used to obtain a first rough estimate. Using this approach, we found evidence of coinfection for 4/336 (1%) episodes with at least two invasive sequences obtained at < 3 days interval. By contrast evidence of co-colonisation was found in 11/167 (7%) episodes with at least two colonising sequences obtained at < 3 days interval. This seems to confirm the prevailing opinion that there is a narrow bottleneck at invasion, however more subtle coinfections are missed with this approach and as mentioned above the estimate is heavily dependent on the density of sampling and sequencing depth. We have added the coinfection estimates to the Results section of the manuscript (lines 127-129) and mentioned the issues around the estimation of the bottleneck size in the discussion (lines 567-569 and lines 745-750).

Other comments that must be addressed1. Line 78: Reference 13 is a study in E. faecalis, not E. faecium. Chilambi et al., 2020 PNAS examine colonization vs. infection in VREfm.

We have corrected the mistake and added the reference on within-host evolution of VRE (lines 75-77).

2. Lines 143-145: Please define "early stage," "late infection stage," "persistence," and "treatment failure" here. These terms are not defined in the methods or anywhere else in the paper.

We have added a definition for these terms (lines 161-162). In addition, we replace the term “treatment failure” with recurrence. Persistence and recurrence are the commonly used to define microbiological failure in *Staphylococcus aureus* bacteraemia, the main syndrome analysed in our study. When revising the manuscript, we have elected not to use the term “treatment failure” as it is not well defined and may include outcomes that are not necessarily linked to bacterial adaptation.

3. Line 153 should reference Figure 2B.

We have modified the figure reference (line 171).

4. Line 183 should reference Figure 2 —figure supplement 2.

We have carefully reviewed figure references for IS insertions: 1) Figure 2 —figure supplement 5 (Figure 2 —figure supplement 2 in the older version) is referenced at line 220, to illustrate the diversity of IS families involved. 2) Figure 2C is referenced in the following sentence, at lines 220-221, to specify that the burst of IS insertions is visible when considering the accumulation of IS insertions within invasive strains (type I>I) in panel C. 2) We hope that this clarifies how the two IS insertions figures are references in the text.

5. Lines 207-208: How many genes had a homolog in the reference strain? What percentage of the genome was this?

A total of 1,736 genes had a FPR3757 homolog (74%). We have added this information to the result section (lines 254-255).

6. Was the definition of "colonization" vs. "infection" consistent across all studies compiled here? If not this should be acknowledged and discussed.

The categorisation of “colonising” or “invasive” was based the following 2-steps assessment of the metadata (lines 799-803): 1) We first checked if the collection site was unambiguously associated with one of the two categories (e.g. “nose”: colonising; “blood” or “bone”: invasive); 2) For sites that could be associated with both colonisation or infection (e.g. “lung” or “skin”) we considered the available details provided in the publication including the categorisation provided by the authors and the clinical histories, if available. While 2,151 sequences (83%) were unambiguous and could be assessed at the first step, we acknowledge that this second categorisation relies on the authors’ assessment. This is now mentioned in the discussion (lines 739-742).

7. Figure 1: Panels B and D are somewhat confusing. Consider revising panel D, or moving it to the supplement.

Following the reviewers’ suggestions we have modified figure 1: panel C is now a “workflow” figure showing the within-host evolution analysis process. We have replaced panel D with a simplified version of the statistical model we used.

8. Figure 3 and others like it: the triangles corresponding to "2" vs. "3" independent acquisitions do not look noticeably different on the figures.

We have modified the figures to improve the readability of panel C (Figure 3, Figure 3 —figure supplements 6 and 7, Figure 4).

9. Figure 3 —figure supplement 1: what does "ext*?" mean?

This is the SnpEff annotation for a stop_lost&splice_region_variant. We have now added an explanation to the legend of Figure 3 —figure supplement 1 (lines 1137-1138).

10. Line 416: what is the reference base of the definition about adaptation based on a centrality value of more than 0?

A centrality value of 0 means that the episode (node) doesn’t have any connection to other episodes. Since connections are based on the Jaccard index, this means that a node with a centrality value of 0 doesn’t share any mutated genes with other episodes. In other words, within-host mutations for these nodes are only found in non-convergent loci (lines 520-522).

11. Line 712-714 and 722-723: These two sentences could be integrated into one.

We have modified this paragraph as suggested (lines 911-913).

12. In general, the article should be written with greater readability for the generalist audience of eLife.

We agree with the reviewer that readability is important, but this is a very broad criticism and therefore somewhat difficult to address. We have re-read the manuscript carefully with an eye to accessibility to a broad scientific audience. In this vein, we preface each section of the results with a rationale for a particular experiment, we define key terms and try to summarise the particular method(s) being deployed. Examples are at lines 177-184 (brief explanation of the regression analysis of mutation rates), lines 186-195 (rationale for the investigation of genome degradation signatures) and lines 208-209 (definition of the neutrality index); lines 295-299 (rationale and approach to infer functional consequences of substitutions); lines 452-464 (rationale and approach to the epistasis analysis); lines 503-523 (rationale and approach to the adaptation network analysis). We have made every effort to explain our implementation of leading-edge statistical and computational techniques as clearly as possible (examples lines 257-265).

13. Please check the whole article again and modify some mistakes such as Line 240, line 259, line 289, line 784, Figure 3 —figure supplement 5, and so on.

We have revised the manuscript and corrected mistakes.